# Generation of functional liver organoids on combining hepatocytes and cholangiocytes with hepatobiliary connections ex vivo

Naoki Tanimizu [1✉], Norihisa Ichinohe[1], Yasushi Sasaki [2,3], Tohru Itoh [4], Ryo Sudo [5], Tomoko Yamaguchi[6], Takeshi Katsuda [7], Takafumi Ninomiya[8], Takashi Tokino [3], Takahiro Ochiya[6], Atsushi Miyajima[4] & Toshihiro Mitaka[1]

In the liver, the bile canaliculi of hepatocytes are connected to intrahepatic bile ducts lined with cholangiocytes, which remove cytotoxic bile from the liver tissue. Although liver organoids have been reported, it is not clear whether the functional connection between hepatocytes and cholangiocytes is recapitulated in those organoids. Here, we report the generation of a hepatobiliary tubular organoid (HBTO) using mouse hepatocyte progenitors and cholangiocytes. Hepatocytes form the bile canalicular network and secrete metabolites into the canaliculi, which are then transported into the biliary tubular structure. Hepatocytes in HBTO acquire and maintain metabolic functions including albumin secretion and cytochrome P450 activities, over the long term. In this study, we establish functional liver tissue incorporating a bile drainage system ex vivo. HBTO enable us to reproduce the transport of hepatocyte metabolites in liver tissue, and to investigate the way in which the two types of epithelial cells establish functional connections.

[1] Department of Tissue Development and Regeneration, Research Institute for Frontier Medicine, Sapporo Medical University School of Medicine, Sapporo, Japan. [2] Biology Division, Department of Liberal Arts and Sciences, Center for Medical Education, Sapporo Medical University, Sapporo, Japan. [3] Department of Medical Genome Sciences, Research Institute for Frontier Medicine, Sapporo Medical University School of Medicine, Sapporo, Japan. [4] Laboratory of Stem Cell Therapy, The Institute for Quantitative Biosciences, The University of Tokyo, Tokyo, Japan. [5] Department of System Design Engineering, Keio University, Yokohama, Japan. [6] Department of Molecular and Cellular Medicine, Institute of Medical Science, Tokyo Medical University, Tokyo, Japan. [7] University of Pennsylvania Perelman School of Medicine, Philadelphia, PA, USA. [8] Department of Anatomy, Sapporo Medical University School of Medicine, Sapporo, Japan. ✉email: tanimizu@sapmed.ac.jp

Epithelial organs consist of multiple types of epithelial tissue, such as alveoli and trachea in the lung, urinary tubules and collecting ducts in the kidney, acini and pancreatic ducts in the pancreas, and bile canaliculi (BC) and bile ducts (BDs) in the liver. The structures connecting two types of tissues are unique to each organ. Various substances, including air, urine, digestive enzymes, and bile are altered in composition as they flow through the connecting structures. It is therefore crucial to accurately connect tissue structures composed of different types of epithelial cells, in order to reproduce the function of each organ ex vivo.

To generate organoids or mini-organs, tissue stem cells are cultured in three dimensions (3D), to enable the cells to grow, divide, and self-organize into tissue structures similar to those found in vivo. This approach has been used to generate functional components of the gastrointestinal tract, including intestinal[1], gastric[2], colonic[3] and hepatic tissues[4,5]. The organoids are likely to contain niches for maintaining tissue stem/progenitor cells, given that these organoids can expand in size. However, each organoid consists of a single type of tissue, and they do not contain structures connecting different types of epithelial cells.

Co-culturing is a technique used to generate organoids consisting of multiple types of tissues. A liver organoid equipped with a vascular system has been developed in a co-culture of hepatoblast-like cells, vascular endothelial cells, and mesenchymal cells, derived from human-induced pluripotent stem cells (hiPSCs)[6]. The hepatocytes in the organoid may exchange metabolic substances via the blood flow when they are transplanted into immunodeficient mice[7]. However, the organoid does not contain the biliary structure that is essential to drain cytotoxic bile from the hepatic tissue.

In this study, we present a mouse hepatobiliary organoid with hepatocyte clusters derived from small hepatocytes (SHs), and a biliary network derived from epithelial cell adhesion molecule positive (EpCAM$^+$) cholangiocytes. SHs are intrinsic hepatocyte progenitors isolated from healthy adult mice as hepatocytes with small size, which subsequently proliferate and differentiate into functional hepatocytes in vitro and in vivo[8,9]. In our hepatobiliary organoids, bilirubin and fluorescein-labeled bile acid were absorbed by the hepatocytes, excreted into BCs, and then accumulated in the biliary system, indicating that a connection between hepatocytes and cholangiocytes had been established. Since hepatocyte clusters are functionally connected to the biliary tubules, we call this organoid a "hepatobiliary tubular organoid (HBTO)". Hepatocytes in an HBTO acquire and maintain metabolic functions for more than one month. The development of HBTOs enables us to investigate the transport of hepatocyte metabolites within the liver tissue and to monitor the metabolism of the hepatocytes in the long-term ex vivo.

## Results

### Induction of hepatobiliary connections in co-cultures of hepatocytes and cholangiocytes.
In order to reconstruct hepatobiliary tissue structures in vitro, we used SHs and primary cholangiocytes isolated from healthy adult mice. SHs proliferate and differentiate into mature hepatocytes (MHs) in tissue culture dishes in the presence of Matrigel (MG) (Supplementary Fig. 1a)[8,9]. SHs form BC-like structures, although they are not organized into a continuous luminal network (arrowheads in Supplementary Fig. 1a). Primary cholangiocytes isolated as EpCAM$^+$ cells proliferate on type I collagen gel and form a tubular network with an overlay of collagen gel (Supplementary Fig. 1b)[10,11].

An approach often used to generate multicellular tissue structures, is spheroid or 3D cultures, in which cell aggregates are embedded in MG[12]. Although hepatocytes generate bile canaliculi-like structures in spheroids[4], biliary networks are not

generated in this type of 3D culture, but they do develop in sandwich culture[10,13]. Therefore, we used a sandwich culture for co-culturing SHs and cholangiocytes. Type I collagen gel was used as the bottom layer, and type I collagen gel containing 20% MG (Col-MG) was used as the top layer. We mixed cholangiocytes and SHs and plated them onto type I collagen gel. Although the cholangiocytes and SHs formed colonies, they did not contact each other, and thus did not form hepatobiliary connections. To establish contact between SHs and cholangiocytes, we added SHs to the culture after the cholangiocytes had proliferated to form colonies on the collagen gel (Fig. 1a). Under these conditions, SHs attached in empty spaces among cholangiocyte colonies, and close contact with cholangiocytes could be observed one day after plating the SHs (left panels of Fig. 1b). The HNF4α$^+$ hepatocytes established tight junctions with the cholangiocytes (Supplementary Fig. 2). Morphogenesis was induced by an overlay of Col-MG. Two weeks after the overlay, hepatocytes showed cellular morphology similar to that of MHs, having a large amount of cytoplasm and round nuclei, and formed a BC network. On the other hand, cholangiocytes formed a tubular network. At this stage, we identified luminal connections between hepatocytes and cholangiocytes under a phase-contrast microscope (right panels of Fig. 1b). Immunofluorescence analysis was performed four weeks after the Col-MG overlay. Serial X–Y images were acquired on a confocal microscope to examine optical cross-sections of the hepatobiliary connections (Fig. 1c). It was apparent that a BC network formed within the HNF4α$^+$ hepatocyte cluster (arrowheads in the upper right panels of Fig. 1c) was connected to a lumen consisting of CK19$^+$ cholangiocytes and HNF4α$^+$ hepatocytes (arrowheads in the middle right panels of Fig. 1c), and then eventually to the duct composed of CK19$^+$ cholangiocytes (arrowheads in the lower right panels of Fig. 1c). In each experiment, using a phase-contrast microscope we could observe hepatobiliary connections on the boundary between the cholangiocytes and the hepatocytes. We evaluated the efficiency of the establishment of hepatobiliary connections in the organoids by staining them with phalloidin and counting the number of lumina connecting the hepatocyte clusters and biliary tubules (Supplementary Fig. 3). We found an average of $3.2 \pm 0.4$ connections (mean ± SEM) per 1 mm of the boundary between hepatocytes and cholangiocytes. Since the hepatocyte clusters were connected to the biliary tubules, we called this organoid an "HBTO".

### Hepatocytes and cholangiocytes maintain their lineages to establish HBTOs.
Recent research has shown that hepatocytes and cholangiocytes are plastic[14–17]. The results shown in Fig. 1c suggest that hepatocytes and cholangiocytes, but not any intermediate cells, form the hepatobiliary junctions. We further examined whether hepatocytes or cholangiocytes show intermediate characteristics, expressing both markers, to form the hepatobiliary junctions. Antibodies against carcinoembryonic antigen-related cell adhesion molecule (CEACAM) were used to specifically label the apical membranes of hepatocytes, and ezrin (EZN) was used to label cholangiocytes. As shown in the upper panels of Fig. 2a, CEACAM$^+$EZN$^-$ hepatocytes (closed arrowhead) and CEACAM$^-$EZN$^+$ cholangiocytes (open arrowhead) formed the junction. We did not find CEACAM$^+$EZN$^+$ cells in the HBTOs. We also investigated the expression of CK19, Sry-HMG box 9 (SOX9), and osteopontin (OPN) as cholangiocyte markers, and radixin (RDX) as a hepatocyte marker (middle and lower panels of Fig. 2a and Supplementary Fig. 4). The apical membranes of hepatocytes (closed arrowheads) and those of cholangiocytes (open arrowheads) surrounded the luminal space at hepatobiliary junctions. We did not find cells expressing both

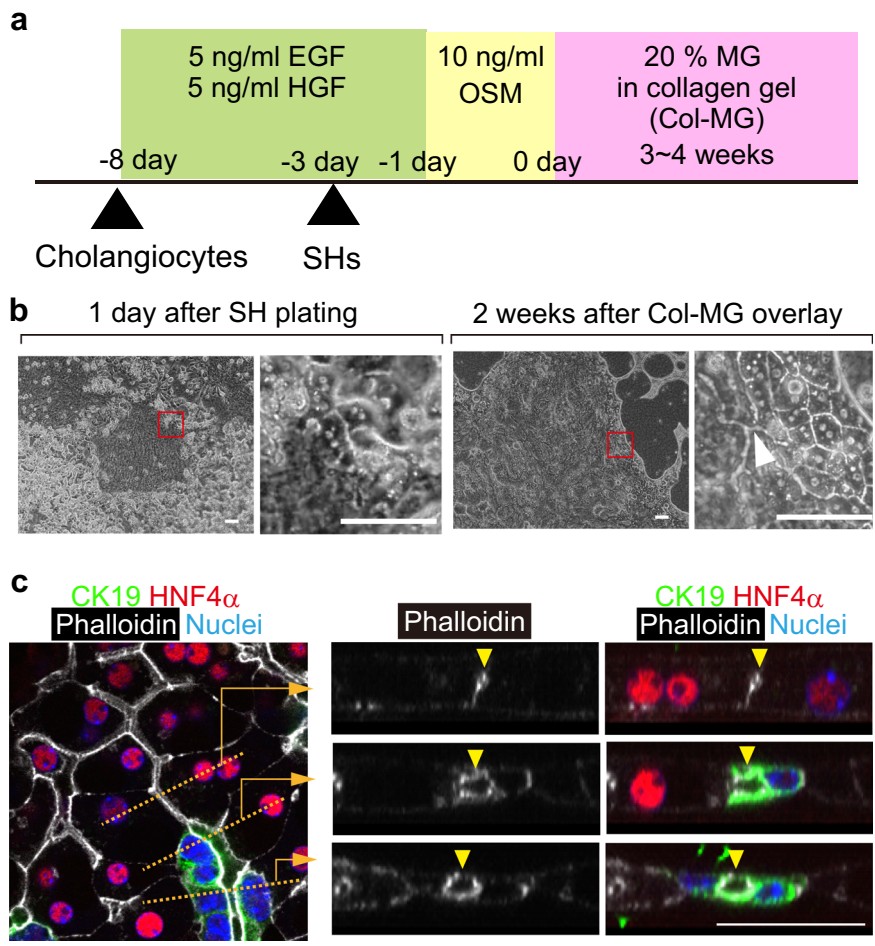

**Fig. 1 Induction of hepatobiliary connections in co-cultures of SHs and cholangiocytes. a** Schedule of co-culture of SHs and cholangiocytes.
**b** Morphological changes in hepatocytes and cholangiocytes in co-culture. Cholangiocytes are in close contact with hepatocytes one day after plating SHs.
At two weeks after the overlay of collagen-containing 20% Matrigel (Col-MG), the BC network of hepatocytes connects to the biliary network (an
arrowhead). Co-culture was repeated more than fifty times. The contacts and connections between hepatocytes and cholangiocytes were always observed
at similar time points. Boxes are enlarged in square panels. Bars represent 100 μm. **c** Hepatobiliary connections in co-culture Optical cross-sections along
broken lines in the left panel are shown in the right panels. The BC within the HNF4α+ hepatocyte clusters (arrowheads in upper right panels) connect to
the lumen, which consists of CK19+ cholangiocytes and HNF4α+ hepatocytes (arrowheads in middle right panels), and then eventually reach the duct of
CK19+ cholangiocytes (arrowheads in lower right panels). At four weeks after MG overlay, the immunostaining with phalloidin (white), anti-HNF4α
antibody (red), anti-CK19 antibody (green), and Hoechst 33342 (blue) was repeated five times independently. Three fields were examined in each sample.
Three out of fifteen areas were further used to collect serial optical sections for 3D reconstruction. The representative images are shown in this figure. Bars
represent 50 μm.

hepatocyte and cholangiocyte markers. As with the hepatobiliary connections observed in HBTO, CEACAM+EZN− hepatocytes and CEACAM−EZN+ cholangiocytes surrounded the lumen at the hepatobiliary junction in adult liver tissue (Supplementary Fig. 5). Three-dimensional analyses were performed using immunofluorescence images of HBTOs with anti-CEACAM and anti-EZN antibodies to reveal the way in which hepatocytes and cholangiocytes formed the hepatobiliary connections in HBTOs. About 60% of the connections consisted of one hepatocyte and one cholangiocyte, whereas others consisted of one hepatocyte and two or three cholangiocytes (Supplementary Fig. 6).

There remained a possibility that hepatocytes were completely converted to cholangiocytes, or vice versa, during the generation of hepatobiliary connections. To examine whether lineage conversion was involved in hepatobiliary morphogenesis, we cultured SHs isolated from CAG-Cre:ROSA-tdTomato mice with wild-type EpCAM+ cholangiocytes (Fig. 2b). The BC of tdTomato+CK19− hepatocytes (the upper far right panel) was

connected to the lumen of tdTomato−CK19+ biliary tissue (the lower far right panel). At the boundary, tdTomato+CK19− hepatocytes and tdTomato−CK19+ cholangiocytes surrounded the luminal structure (the middle far right panel). We did not detect tdTomato+ cholangiocytes or tdTomato− hepatocytes. We also cultured EpCAM+ cholangiocytes isolated from CAG-Cre: ROSA-tdTomato mice with SHs isolated from wild-type mice (Supplementary Fig. 7). In this culture, the BC network of hepatocytes (closed arrowheads in Supplementary Fig. 7a) was connected to the biliary network of tdTomato+ cholangiocytes (open arrowheads in Supplementary Fig. 7a). Some tdTomato+ cholangiocytes were weakly positive for HNF4α, although their cellular morphology was visually similar to that of the neighboring cholangiocytes (Supplementary Fig. 7b). Neither tdTomato+ hepatocytes nor tdTomato− cholangiocytes were detected. Collectively, hepatocytes and cholangiocytes maintained their original lineages and formed hepatobiliary connections in HBTOs (Supplementary Table 1).

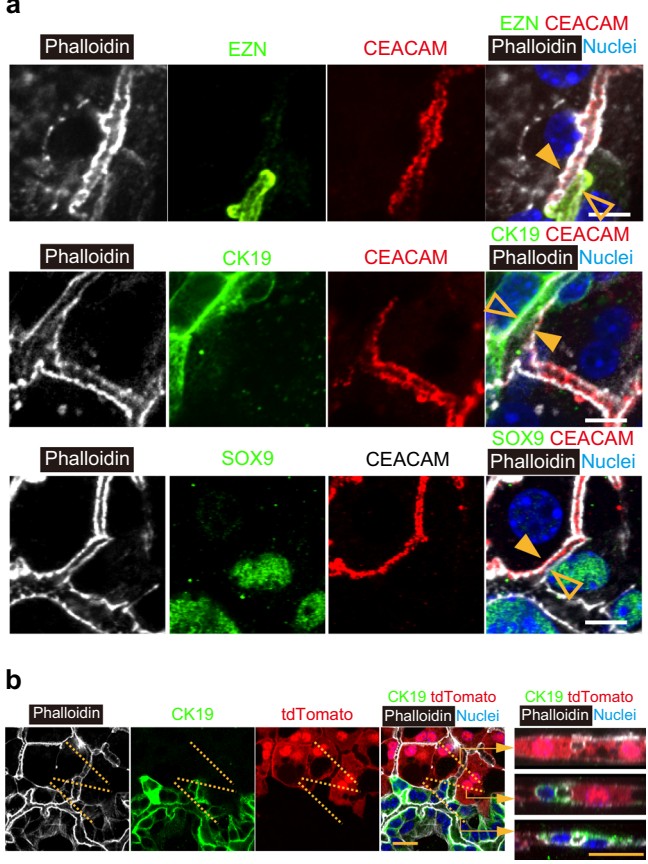

**Fig. 2 Hepatocytes and cholangiocytes maintain their lineage to establish hepatobiliary junctions. a** Immunological identification of the hepatobiliary junction. The hepatobiliary junction consists of hepatocytes (closed arrowheads) and cholangiocytes (open arrowheads). The continuous lumen stained with phalloidin (white) is surrounded by CEACAM+EZN− hepatocytes and CEACAM−EZN+ cholangiocytes, CEACAM+CK19− hepatocytes, and CEACAM−CK19+ cholangiocytes, or CEACAM+SOX9− hepatocytes and CEACAM−SOX9+ cholangiocytes. The immunostaining using phalloidin, anti-CEACAM antibody (red), and Hoechst 33342 (blue) with anti-EZN (green), anti-CK19 (green), or anti-SOX9 (green) antibody was repeated three times independently. Three fields were examined in each sample and the representative images are shown in this figure. Bars represent 20 μm. **b** tdTomato+ hepatocytes maintain their lineage to form a continuous luminal network with cholangiocytes. Optical cross-sections along broken lines in the square four panels are shown in the far right panels. The luminal network among tdTomato+ hepatocytes (upper right panel) is connected to that of the tdTomato−CK19+ biliary structure (lower right panel) at the boundary (middle right panel). The luminal network is recognized by F-actin bundles visualized using phalloidin. The immunostaining with phalloidin (white), anti-CK19 antibody (green), and Hoechst 33342 (blue) was repeated three times, independently. Two fields were examined in each sample. Two out of six areas were further used to collect serial optical sections for 3D reconstruction. The representative images are shown in this figure. Bars represent 40 μm.

**Transport of hepatocyte metabolites in HBTOs**. A variety of metabolic reactions occur within functional hepatocytes. The metabolites produced by these reactions are secreted into the BC and then transported into intrahepatic bile ducts (IHBDs). To demonstrate functional connections between hepatocytes and cholangiocytes in HBTOs, we investigated whether the metabolites produced by hepatocytes were secreted into the BC and eventually transported into tubular structures consisting of cholangiocytes. We applied chloromethylfluorescein diacetate (CMFDA) to HBTOs and examined whether the compound accumulated in the biliary tissue. Hepatocytes, but not cholangiocytes, incorporated and degraded CMFDA to produce fluorescein, which was secreted into the apical luminal space (Supplementary Fig. 8). Fluorescein accumulated in the biliary tissue when hepatocytes and cholangiocytes established functional connections. Fluorescein was detected at high levels in the BC network 10 min after the incubation of HBTOs in CMFDA-containing medium (Fig. 3a). Although some fluorescein was observed in the biliary networks near the hepatocyte clusters at this time, more fluorescein was found to have been transported to the biliary tissue after 120 min of incubation. We then added a fluorescein-labeled bile acid, cholyl-lysine fluorescein (CLF), to HBTOs, and followed its transport. CLF was incorporated into hepatocytes, but not into the cholangiocytes (Supplementary Fig. 8). CLF was then secreted to the BC through the bile salt efflux pump. CLF had been secreted to the BC by 30 min after the treatment and had accumulated in the biliary tissue after 6 h (Fig. 3b and Supplementary Fig. 9), indicating that the CLF secreted into the BC was transported to the biliary network. Finally, HBTOs were exposed to bilirubin, and we investigated whether the bilirubin was metabolized and then accumulated in the biliary network. After five days of incubation, bilirubin in the luminal spaces of the HBTO was visualized by oxidizing the bilirubin to biliverdin. Biliverdin was detected in both the biliary and the BC networks (Fig. 3c), indicating that the hepatocytes took up and modified bilirubin, which was then excreted to the BC and transported into the biliary network. These results indicate that the HBTOs reproduced the transport of hepatocyte metabolites in liver tissue in vivo.

**Hepatocytes maintain metabolic functions in HBTOs**. In order to provide an assay system for drug metabolism, it is important to induce the cellular characteristics of well-differentiated hepatocytes in the HBTOs. Hepatocytes in an HBTO could secrete three times as much ALB as SH-derived hepatocytes (Hep) (Fig. 4a). HBTOs showed higher CYP1A1-like and CYP3A4-like activities when compared with Hep (Fig. 4b). This CYP3A4-like activity is comparable to that shown by MHs in culture (Supplementary Fig. 10). The secretion of ALB by HBTOs gradually increased, and a high level of ALB secretion persisted for more than a month (Fig. 4c). Quantitative PCR data pertaining to *Cyp* expression (Supplementary Fig. 11c) and immunofluorescence detecting ALB and CYP3A (Supplementary Fig. 12) indicated that CYP activity and ALB secretion were associated with hepatocytes, but not cholangiocytes, in HBTOs. Hepatocytes in HBTOs also absorbed DiI-AcLDL, a further indication that they were functional (Fig. 4d). The generation of HBTOs promoted hepatic functions and contributed to maintaining these functions for the long term. We consider that the formation of hepatobiliary connections is correlated with improvement of hepatocyte functions, although we cannot exclude a possibility that co-culture conditions may favor the maturation of hepatocytes in HBTOs.

**Cholangiocytes maintain secretory functions in the hepatobiliary organoid**. Quantitative PCR analysis showed that cholangiocyte markers were expressed in HBTOs (Supplementary Fig. 13a). Immunofluorescence analysis indicated that the cholangiocyte markers, including EZN, CK19, SOX9, and OPN, were expressed in the cholangiocytes comprising HBTOs (Fig. 2 and Supplementary Fig. 4). Rhodamine 123 loaded onto HBTOs four weeks after Col-MG overlay was incorporated into the biliary structure, depending on MDR activity (Supplementary Fig. 13b). In response to forskolin, which increases the level of cAMP, the

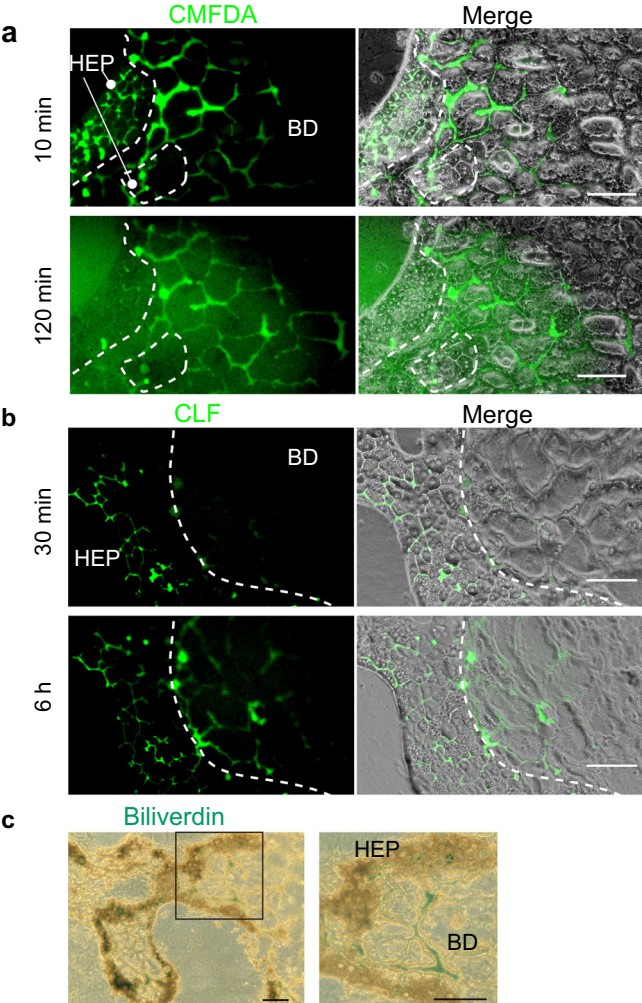

**Fig. 3 Transport of hepatocyte metabolites in HBTO. a** Transport of fluorescein diacetate in HBTOs. Fluorescein derived from chloromethyl fluorescein diacetate (CMFDA) is secreted into BCs and then transported into the biliary network. After 10 min of incubation, fluorescein is strongly detected in the BC network of the hepatocyte cluster (HEP, surrounded by a broken line), as well as in the biliary network (BD). Fluorescein is absent from the hepatocyte clusters and reaches the biliary network after 120 min incubation. HBTOs were incubated in the presence of CMFDA for 10 min. Phase-contrast and fluorescent images were taken after washing with culture medium five times. HBTOs were kept in the culture medium without CMFDA. Dotted lines indicate the boundary between HEP and BD. The experiment on one well was repeated three times, independently. Two fields were examined in each well and the representative images are shown in this figure. Bars represent 100 μm. **b** Transport of bile acids in HBTO. CLF taken up by hepatocytes is transported into the biliary network. The organoids were incubated in the presence of CLF for 30 min. After five washes, images were taken at 30 min (panels 1 and 2) and at 6 hours (panels 3 and 4). Broken lines indicate the boundary between HEP and BD. The experiment using one well was repeated three times, independently. Three fields were examined in each well and the representative images are shown in this figure. Bars represent 100 μm. **c** Transport of bilirubin. Bilirubin taken up by hepatocytes is excreted into BCs and then transported into the biliary network. Bilirubin in HBTOs was visualized by turning it into biliverdin. The experiment on one well was repeated three times, independently. Three fields were examined in each well and the representative images are shown in this figure. The box in the left panel is enlarged in the right panel. Bars represent 200 μm.

luminal spaces of the biliary network were expanded (Supplementary Fig. 13c). We also analyzed biliary structures using a transmission electron microscope and found microvilli but no primary cilia on the apical surface (Supplementary Fig. 14). These results indicate that cholangiocytes were functional in the HBTOs.

**SHs is a subfraction of periportal and centrilobular hepatocytes.** We established HBTOs using cholangiocytes and SHs. We tested MHs as a source of hepatocytes for HBTOs, but they did not form connections with cholangiocytes as efficiently as SHs (Fig. 5a). As we previously reported[8,9], SHs express typical hepatocyte markers, including *Hnf4a*, *Cps1*, and *Tdo2* (Supplementary Fig. 15a). In order to further clarify the differences between SHs and MHs, gene expression profiles were analyzed using RNA-seq. The two types of hepatocytes showed very similar gene expression profiles, but the expression of *Cyps* was lower in SHs than in MHs (Supplementary Fig. 15b). Genes related to the Wnt signaling pathway were expressed at a lower level in SHs than in MHs (Supplementary Fig. 15c). Hepatocytes can be categorized into three fractions—ZONE1, ZONE2, and ZONE3—depending on their localization on the tissue along the portal vein (PV) to the central vein (CV) axis[18]. Using this categorization, the WNT/β-catenin signal was active in hepatocytes in ZONE3[19,20]. Quantitative PCR analysis further demonstrated that WNT target genes such as *Gs*, *Lgr5*, and *Axin2*, which are strongly expressed in ZONE3 hepatocytes, were only weakly expressed in SHs (Supplementary Fig. 15c). *Cyp3a11* and *1a2* are also highly expressed in ZONE3 hepatocytes[20]. These data strongly suggest that SHs are a subfraction of ZONE1 and ZONE2 hepatocytes. To investigate any correlation between the localization of SHs and their capability for establishing hepatobiliary connections with cholangiocytes, we tried to establish a protocol separating hepatocytes in ZONE1 and ZONE2 from hepatocytes in ZONE3 and applied fractionated hepatocytes to an HBTO culture.

Membrane proteins are useful for distinguishing between the different cellular populations. In the liver, E-cadherin (ECAD) is expressed in ZONE1 and ZONE2 hepatocytes, and claudin-2 (CLDN2) is expressed in ZONE3 hepatocytes (Fig. 5b). Consistent with the RNA sequence data, which indicated that SHs were in ZONE1 and ZONE2, SHs expressed more *Ecad*/*Cdh1* and less *Cldn2* than MHs (Fig. 5c). FACS analysis further demonstrated that SHs are strongly positive for ECAD, whereas MHs contain both ECAD+ and ECAD− cells (Supplementary Fig. 16 and Fig. 5d). ECAD+ MHs generated hepatobiliary connections with cholangiocytes more efficiently than did ECAD− MHs (Fig. 5e). These results indicate that MHs in ZONE1 and ZONE 2, and SHs, which share the cellular characteristics of strong ECAD expression, can be involved in hepatobiliary connections with cholangiocytes.

**Generation of hybrid HBTOs containing human hepatocytes.** For pharmaceutical and clinical applications, human hepatocytes must be introduced into liver organoids. To this end, we co-cultured human reprogramming hepatocytes (hCLiP), which can robustly differentiate to form functional hepatocytes in vitro[21], with mouse cholangiocytes. When hCLiP were cultured alone, they differentiated into hepatocytes morphologically similar to MHs but did not form BC-like structures (two left panels of Fig. 6a). In co-culture with tdTomato+ mouse cholangiocytes, BC-like structures were evident (two right panels of Fig. 6a). Immunofluorescence analysis demonstrated that the BC-like structure (the upper right panels of Fig. 6b) was connected to the biliary structure (the lower right panels) via a lumen

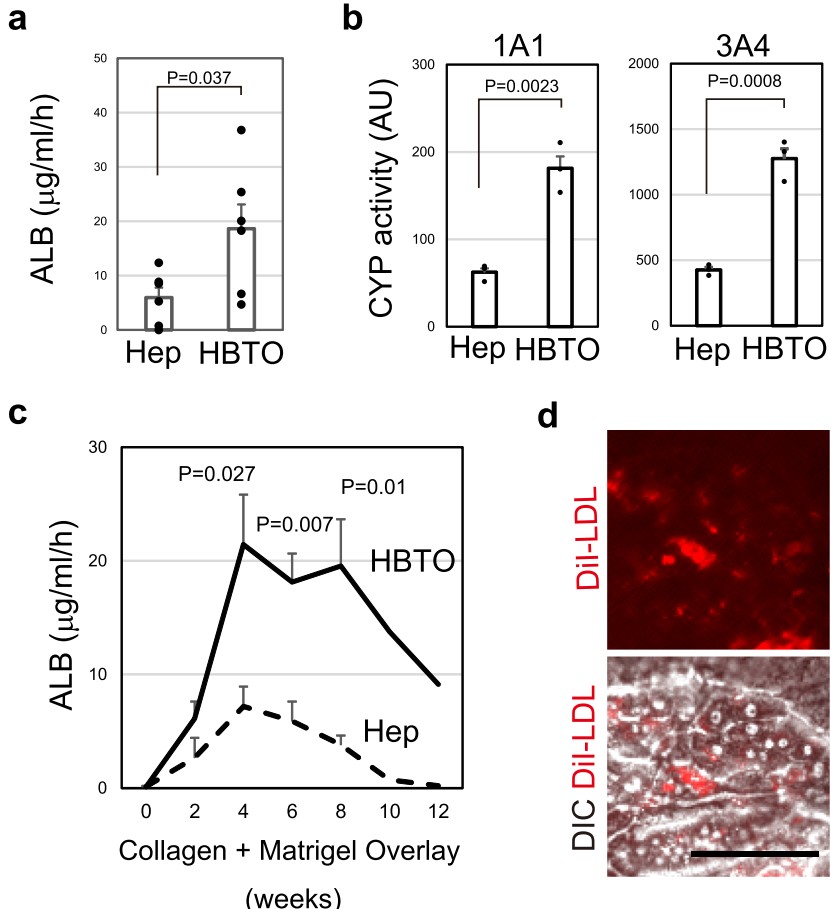

**Fig. 4 Hepatocytes maintain metabolic functions in HBTO. a** ALB secretion from hepatocytes and HBTOs. Hepatocytes in HBTOs secrete more ALB than those cultured alone. SHs were plated onto collagen gel with or without cholangiocyte colonies. Culture medium was collected, and ALB concentration was examined by ELISA four weeks after Col-MG overlay. Six wells ($n = 6$) cultured independently were used to determine ALB concentration in the culture medium. Error bars represent SEM. Two-tailed unpaired $t$-tests were performed to compare ALB secretion between HBTO and SHs. **b** Activity of cytochrome P450s in hepatocytes and HBTOs. Hepatocytes in HBTOs show higher levels of CYP1A1, and CYP3A4-like activity than those cultured alone. CYP activities were examined three weeks after Col-MG overlay. The average values of CYP activity measured for three different culture wells ($n = 3$) are shown in the graphs. Assays for the same number of wells were repeated two and three times, independently, for CYP1A1 and CYP3A4, respectively. Error bars represent SEM. Two-tailed unpaired $t$-tests were performed to compare CYP activities between HBTO and SHs. **c** Long-term maintenance of ALB secretion. ALB secretion gradually increases for four weeks after Col-MG overlay. Then, hepatocytes in HBTO maintain ALB secretion at a similar level for the following three to four weeks. ALB secretion was observed for two months for five independent cultures ($n = 5$); average values with SEM at each time point during a two-month period are shown in this figure. Two culture wells were further extended to measure ALB secretion at ten and twelve weeks after Col-MG overlay. Two-tailed unpaired $t$-tests were performed to compare ALB secretion between HBTO and SHs at 4, 6, and 8 weeks after MG-Col gel overlay, **d** Hepatocytes in HBTOs take up low-density lipoprotein (LDL). Hepatocytes in HBTOs can take up DiI-AcLDL in supplemented culture medium. HBTOs were maintained for four weeks and further incubated in the presence of DiI-AcLDL for 1 h. The experiment was repeated twice independently. Three fields were examined in each sample and the representative images are shown in this figure. A bar represents 100 μm.

consisting of hepatocytes and cholangiocytes (the middle right panels). The efficiency of the formation of hepatobiliary connections between hCLiPs and mouse cholangiocytes was 2.1 ± 0.1 per 1 mm of the boundary (Supplementary Fig. 17). When CLF was added to the medium, it accumulated in the luminal space, which consisted of tdTomato+ cholangiocytes (Fig. 6c), indicating that human hepatocytes and mouse cholangiocytes can establish functional connections.

**Discussion**
To maintain healthy functional hepatocytes in vitro or in vivo, liver tissue must be equipped with a bile excretion system. In this report, we present HBTO, a hepatobiliary organoid, in which the BC network of hepatocytes connects to the biliary network. The hepatobiliary connection greatly contributes to the long-term maintenance of functional hepatocytes.

To reproduce in vivo liver functions, in particular, the transport of metabolites produced by hepatocytes, the liver tissue architecture should be implemented ex vivo. Liver organoids containing hepatocytes and cholangiocytes have been generated by applying a combination of cytokines and growth factors, mimicking the developmental processes, to cultures of iPS, ES cells, or fetal liver stem/progenitors. However, the functions of these hepatocytes are not comparable to those of MHs, and the formation of the hepatobiliary connection remains unclear[22,23]. Among those challenges, Ramli et al. established a liver organoid associated with biliary cysts[24]. They demonstrated that bile acid analog taken up by induced hepatocytes eventually accumulated in the cystic structures, although it is not clear how BCs and biliary lumen are connected. It is still difficult to confer mature hepatic functions on induced hepatocytes. In contrast to previous reports[22-24], we used cholangiocytes and committed progenitors

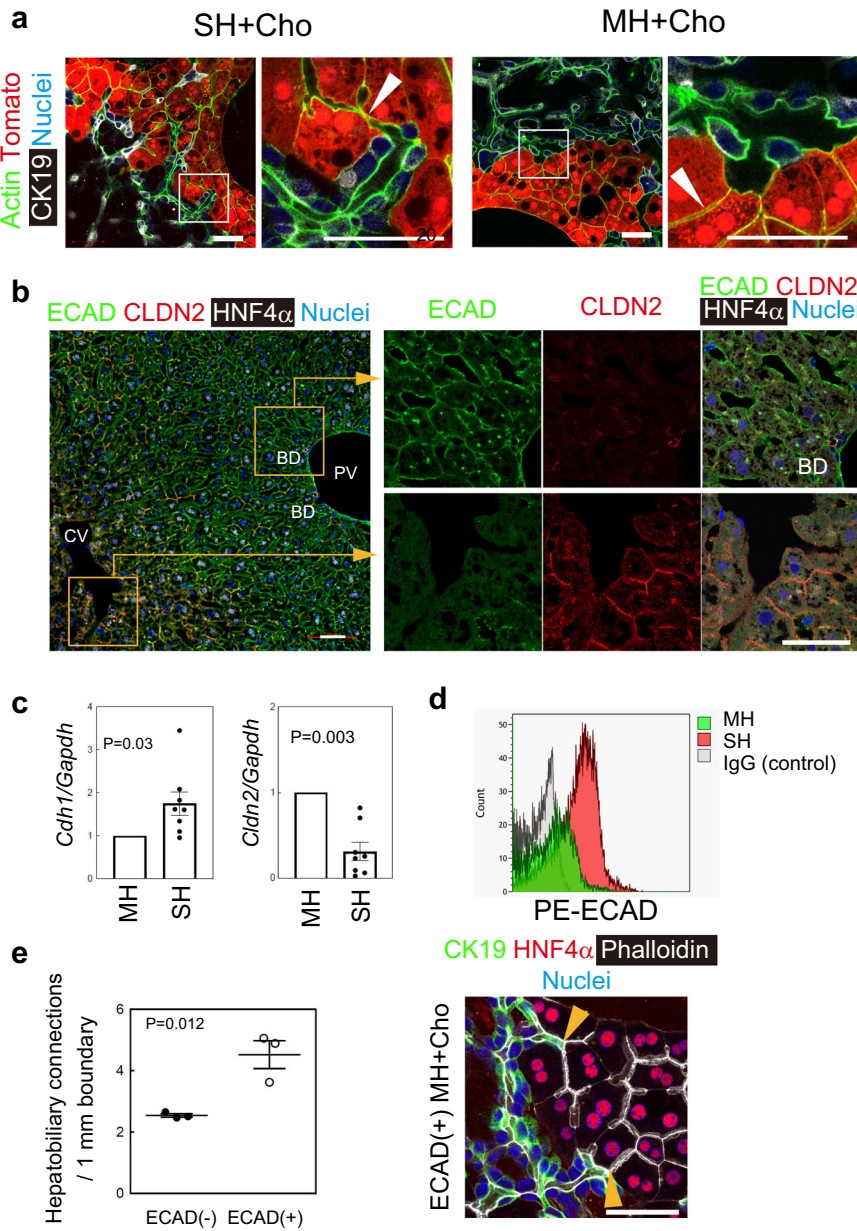

**Fig. 5 ECAD⁺ hepatocytes form hepatobiliary junctions with cholangiocytes. a**. SHs, but not MHs, efficiently form hepatobiliary connections with cholangiocytes at their boundaries. Both SHs and MHs form BCs (arrowheads) in coculture with cholangiocytes (Cho). However, the BCs of tdTomato⁺ SHs connect to CK19⁺ BD, whereas BCs of MHs do not connect to BD. The immunostaining with phalloidin (green), anti-CK19 antibody (white), and Hoechst 33342 (blue) was repeated three times independently. Three fields were examined in each sample and the representative images are shown in this figure. Boxes in left-side panels of SH + Cho and MH + Cho are enlarged in the right-side panels, respectively. Bars represent 50 μm. **b**. ECAD is expressed in ZONE1&2 and CLDN2 in ZONE3 hepatocytes in the adult liver. Hepatocytes in ZONE1&2 near the PV are ECAD⁺CLDN2⁻ (upper right panels), whereas those in ZONE3 near the CV are ECAD⁻CLDN2⁺ (lower right panels). An adult liver section was stained with anti-ECAD (green), anti-CLDN2 (red), and anti-HNF4α (white) antibodies. Nuclei were counterstained with Hoechst 33342 (blue). The immunostaining with anti-ECAD, anti-CLDN2, HNF4α antibodies, and Hoechst 33342 was repeated on sections prepared from two different mice. Three fields were examined in each section and the representative images are shown in this figure. Boxes in the left panel are enlarged in the right panels. Bars represent 50 μm. **c**. SHs express more *Cdh1* and less *Cldn2* than MHs. *Cdh1/Ecad* is higher in SHs than in MHs. qPCR analysis shows that *Cldn2* is expressed at a lower level in SHs than in MHs. MHs and SHs were isolated from the same mouse eight times independently and used for qPCR analysis ($n = 8$). Bars represent SEM. Two-tailed paired *t*-tests were performed to compare gene expression between MHs and SHs. **d**. ECAD expression patterns are different in SHs and MHs. FACS analysis shows that all SHs are ECAD⁺, whereas MHs consist of both ECAD⁻ and ECAD⁺ cells. MHs and SHs were incubated with PE-conjugated anti-ECAD antibody, and ECAD expression was examined using a FACSAria equipped with a green laser. The FACS plot is representative of two independent analyses. The gating for analysis of ECAD expression in SHs and MHs is shown in Supplementary Fig. 16. **e**. ECAD⁺ MHs form hepatobiliary junctions. ECAD⁺ MHs form hepatobiliary junctions with cholangiocytes more efficiently than ECAD⁻ MHs. Co-culture and immunostaining with anti-CK19 antibody (green), anti-HNF4α antibody (red), phalloidin (white), and Hoechst 33342 (blue) were repeated three times ($n = 3$). Each dot in the graph represents the average value of hepatobiliary connections counted in three to four areas in each culture sample. Unpaired two-tailed *t*-tests were performed using Microsoft Excel. Bars represent SEM. The right panel shows the representative image of three independent experiments. Scale bar represents 50 μm.

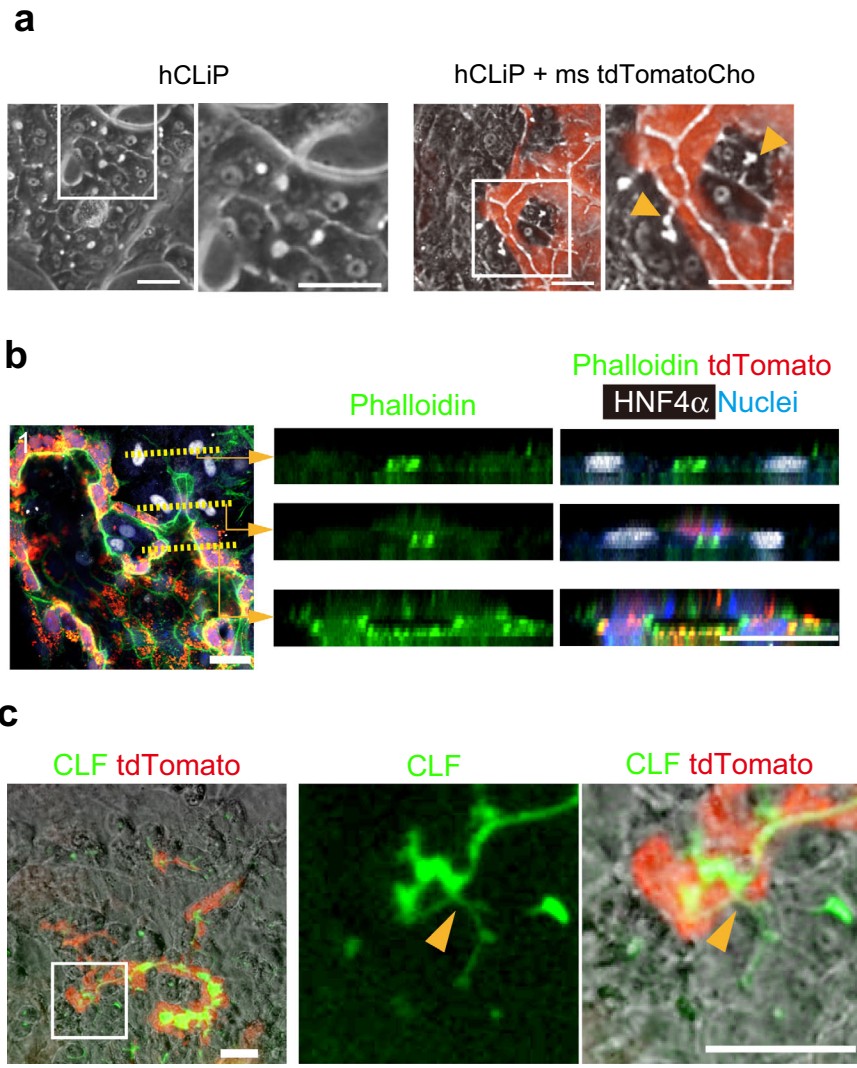

**Fig. 6 Human hepatocytes and mouse cholangiocytes form hybrid HBTOs. a** hCLiP forms bile canaliculi in co-culture with cholangiocytes. Bile canaliculi formed in co-culture are indicated by arrowheads. hCLiP and mouse tdTomato[+] cholangiocytes (Cho) were co-cultured to induce human-mouse hybrid HBTOs. Coculture was repeated four times. Three fields were examined in each well and the representative images are shown in this figure. Boxes are enlarged in the neighboring panels. Bars represent 40 μm. **b** Hepatobiliary junction in a hybrid HBTO. Optical cross-sections along broken lines in the left panel are shown in the right panels. The bile canaliculi in an HNF4α[+] hepatocyte cluster (upper right panels) is connected to the lumen of a tdTomato[+] mouse biliary structure (lower right panels) via the luminal structure surrounded by HNF4α[+] hepatocytes and tdTomato[+] mouse cholangiocytes (middle right panels). The immunostaining with phalloidin (green), anti-HNF4α antibody (white), and Hoechst 33342 (blue) was repeated four times independently. Four fields were examined in each sample. Three out of sixteen filed were further analyzed to collect serial optical sections for 3D reconstruction. The representative images are shown in this figure. Bars represent 20 μm. **c** Hybrid HBTOs take up CLF and transport it to the biliary tissue. CLF taken up by hCLiP is transported to the biliary tissue consisting of tdTomato[+] mouse cholangiocytes. The hepatobiliary connection is indicated by arrowheads. Experiments were repeated twice, independently. Four different fields were examined in each sample and the representative images are shown in this figure. The box in the left panel is enlarged in the middle and right panels. Bars represent 50 μm.

for hepatocytes to generate hepatobiliary organoids. Hepatocyte progenitors mature structurally and functionally in HBTOs in a similar way as MHs, and the apical domain of hepatocytes and cholangiocytes cooperatively construct the hepatobiliary connection. Thus, we have successfully constructed a hepatobiliary tissue structure ex vivo and showed that hepatocyte metabolites are transported from the BCs to BDs in HBTOs.

As part of the process of determining the culture conditions, we first cultured SHs on 2, 4, and 6 mg/ml type I collagen gel, and found ALB secretion was highest in culture using 4 mg/ml collagen gel. Therefore, we chose to use 4 mg/ml type I collagen as the bottom layer in HBTO culture. We counted hepatobiliary connections under a phase-contrast microscope and found that the largest number of connections was made when we used a

collagen gel containing 20% MG as the top layer in HBTO culture, compared with collagen or MG alone.

The sequential plating of SHs and cholangiocytes, and the spontaneous induction of BD and BC morphogenesis, are key to establishing HBTOs. We generally analyzed the structures and functions of HBTO three to four weeks after inducing morphogenesis, since at that time the hepatocytes had become functionally mature, as judged by ALB secretion. No cell death in HBTOs was detected three weeks after Col-MG overlay (Supplementary Fig. 18a), but it was induced by hepatotoxin, and the activity of a hepatocyte-specific enzyme, alkaline phosphatase, was detected in the culture medium (Supplementary Fig. 18b). To gain more insight into the way in which hepatobiliary connections are established in HBTOs, we also performed immunofluorescence

analysis one week after Col-MG overlay and found that hepatocytes and cholangiocytes had formed junctional structures by this time (Supplementary Fig. 19a). The nascent BC in the HNF4α$^+$ hepatocyte cluster (a white arrowhead) had already connected to the CK19$^+$ biliary structure (a yellow arrowhead), and the number of connections did not increase significantly beyond one week (Supplementary Fig. 19b). Comparing HBTOs at one week with those at four weeks, it was evident that the BC network continued to extend for at least four weeks (Supplementary Fig. 19b). This process is probably similar to hepatobiliary morphogenesis in vivo. At embryonic day 17 (E17) (Supplementary Fig. 19c), the hepatobiliary connections already exist (yellow arrowheads), although the BC network is still discontinuous (white arrowheads). In the adult liver, the BC eventually forms a continuous network, and the hepatobiliary system is established (Supplementary Fig. 19c)[25]. Therefore, HBTO could be useful for analyzing the process and regulatory mechanisms of hepatobiliary morphogenesis in vitro.

Recent reports have indicated that hepatocytes consist of several fractions of cells: SOX9$^+$ hybrid hepatocytes[26] and Mfsd2a$^+$ exist near the PV[27], whereas AXIN2$^+$ hepatocytes are next to the CV[28]. In addition to these subpopulations, our gene expression analysis suggested that SHs are a subfraction of hepatocytes in ZONE1 and ZONE2 (Supplementary Fig. 15). Currently, the relationship between SHs and the previously reported hepatocyte subpopulations such as SOX9$^+$, Mfsd2a$^+$, and AXIN2$^+$ cells, remains unclear, although SHs express less Axin2 than MHs, consistent with the localization of SHs in ZONE1 and ZONE2, but not in ZONE3.

As is characteristic of hepatocytes in ZONE1 and ZONE2, SHs are ECAD$^+$. This status may be crucial for generating hepatobiliary connections with ECAD$^+$ cholangiocytes (Supplementary Fig. 20), given that cadherin proteins form homophilic interactions. SHs showed a higher proliferative capability than MHs, an ability that may assist in the establishment of numerous contacts with cholangiocytes in co-culture before the induction of hepatobiliary morphogenesis. SHs expressed lower levels of Cyps than MHs, and this characteristic was retained in HBTOs, as judged by the relatively low levels of Cyp3a11 and 2e1 (Supplementary Fig. 11). However, HBTOs derived from SHs showed CYP3A4-like activity comparable to that of primary MHs, suggesting that SHs are mature in HBTOs (Supplementary Fig. 10). Therefore, we consider that SHs are currently the preferred source of hepatocytes for generating HBTOs. We found that epithelial cells of pancreatic ducts establish a continuous luminal network with SHs, although we have not confirmed whether the connection is functional by examining the transport of fluorescent dye or hepatocyte metabolites. Pancreatic duct cells forming colonies on collagen gel were ECAD$^+$ (Supplementary Fig. 21). This result shows a possibility that our culture methods might be applicable to connect a range of different epithelial tissues, other than hepatocyte clusters and biliary tubules, ex vivo.

Liver transplantation is a curative therapy for fatal liver diseases. However, because of a shortage of donors, the development of new therapies is important, and hepatocyte transplantation has been considered as an alternative. Hepatocytes and hepatocyte-like cells derived from intrinsic and extrinsic stem/progenitor cells can repopulate the recipient liver in mouse and rat models[29,30]. In those experimental models, however, the proliferation of residual hepatocytes must be genetically or pharmacologically suppressed, to allow transplanted cells to proliferate and become engrafted within the recipients' liver tissue. Such pharmacological suppression cannot be used in human patients. Liver organoids such as HBTOs may have advantages over hepatocytes; since HBTOs already contain clusters of functional hepatocytes, they do not have to expand in the recipient's liver.

HBTOs also contain a bile excretion system, which helps hepatocytes maintain their functions in the long term, without being affected by excess accumulation of cytotoxic bile. In the future, it will be necessary to explore a method for introducing liver organoids into a recipient's liver and to develop techniques for connecting the biliary tissue in HBTOs to the BDs of recipients, to secure permanent bile drainage.

In this work, we established a hepatobiliary organoid called an HBTO, in which the BC network in hepatocyte clusters is functionally connected to the biliary network. This hepatobiliary organoid paves the way for the generation of a system for directly monitoring the flux of hepatocyte metabolites within the liver tissue ex vivo. By exploring ways to collect hepatocyte metabolites from biliary tissue, an assay system for drug metabolism could be established. HBTOs containing human hepatocytes such as hCLiPs are useful for research into liver injuries and the development of new drugs. Our results provide a basic concept, and a system, for generating functional liver tissue ex vivo.

## Methods

**Mice**. C57BL6 mice were purchased from Sankyo Labo Service Corporation, Inc. (Tokyo, Japan). CAG-Cre:ROSA-LSL-tdTomato mice were obtained by crossing B6.Cg-Tg(CAG-Cre)CZ-MO2Osb (CAG-Cre) mice from Riken BRC[31] with B6.Cg-Gt(ROSA)26Sortm(CAG-tdTomato)Hze/J (ROSA-LSL-tdTomato) mice (The Jackson Laboratory, Bar Harbor, ME). Eight to twelve-week-old mice were used for cell isolation. Mice were maintained in the SPF animal facility of the Sapporo Medical University, where light/dark cycle, ambient temperature, and humidity are maintained 12 h, 23 ± 2 °C, and 40%, respectively. All animal experiments were approved by the Sapporo Medical University Institutional Animal Care and Use Committee and were conducted according to institutional guidelines for ethical animal use.

**Culture materials**. High concentration type I collagen, and growth factor reduced Matrigel (MG) were purchased from Corning (Corning, NY). Epidermal growth factor (EGF) and hepatocyte growth factor (HGF) were purchased from Corning. Oncostatin M (OSM) was purchased from R&D Systems (Minneapolis, MN). Tissue culture plates (24-well) were purchased from Greiner Bio-One (Kremsmünster, Austria). DMEM/F-12 medium (Sigma-Aldrich, St. Louis, MO) supplemented with 10% FBS (MP Biomedicals, Santa Ana, CA), 10 mM nicotinamide (Sigma-Aldrich), $1 \times 10^{-7}$ M dexamethasone (Dex, Sigma-Aldrich), and 1× ITS (Gibco, Grand Island, NY) was used as the basic medium. The growth medium was prepared by adding 5 ng/ml EGF and 5 ng/ml HGF to the basic medium. The differentiation medium was prepared by adding 1% DMSO (Sigma-Aldrich) to the basic medium.

**Cell Isolation**. To isolate hepatocytes and cholangiocytes from adult mouse liver, two-step collagenase perfusion was performed, as previously reported[9,10]. Briefly, a butterfly needle was inserted into the portal vein and then 25 ml of pre-perfusion solution was injected by using a peristaltic pump at 6 ml/min. Then, the liver was perfused with the perfusion solution containing collagenase by using a peristaltic pump at 3 ml/min. MHs and SHs were obtained from digested tissue, and undigested tissue was kept for isolating cholangiocytes. After collagenase perfusion, the cell suspension was centrifuged at $50 \times g$ for 1 min. The pellet was suspended in Hanks' balanced salt solution, mixed with Percoll (Sigma-Aldrich), and centrifuged at $50 \times g$ for 15 min to eliminate dead cells, yielding MHs. The supernatant collected after centrifugation at $50 \times g$ was further centrifuged at $115 \times g$ for 3 min. The pellet was suspended in Hanks' balanced salt solution, mixed with Percoll, and centrifuged at $180 \times g$ for 15 min to eliminate dead cells, yielding SHs. For RNA seq and quantitative PCR analyses, SHs were further purified as CD31$^-$CD45$^-$EpCAM$^-$ICAM-1$^+$ cells by FACSAriaII[9]. The residual tissue after collagenase perfusion was further digested with collagenase/hyaluronidase solution. Liberated cells were used for isolating cholangiocytes, based on the expression of EpCAM, by a magnetic cell sorter (Miltenyi Biotec, Bergisch Gladbach, Germany).

**Induction of the HBTOs**. A step-by-step protocol describing induction of the HBTO can be found at Protocol Exchange[32]. Cholangiocytes were resuspended in the growth medium and plated in 24-well plates coated with 200 μl of 4 mg/ml type I collagen gel prepared from high concentration type I collagen (Corning) at a density of 50,000 cells/well. Five days after plating, SHs were added to each well at a density of 50,000 cells/well. Following a further two days of incubation, the medium was replaced with differentiation medium supplemented with 10 ng/ml OSM and then overlaid with collagen gel containing 20% MG (Col-MG), which was prepared by mixing 2 mg/ml type I collagen gel and MG (v/v = 4:1) on ice. The plate was incubated at 37 °C for 3–4 h to form a gel, and then the differentiation

medium was added. Culture medium was replaced with fresh medium every four days.

**Immunostaining and confocal imaging**. A sandwich culture was fixed in PBS containing 4% paraformaldehyde at 4 °C for 30 min with gentle shaking. After washing with PBS, the samples were permeabilized in PBS containing 1% Triton X-100 at room temperature for 30 min. After blocking in Block ACE (KAC Co., Ltd., Kyoto, Japan) containing 0.1% Triton X-100, the samples were incubated with primary antibodies and then dye-conjugated secondary antibodies were applied. Nuclei were counterstained with Hoechst33342. The primary and secondary antibodies used for immunostaining are listed in Supplementary Tables 2 and 3. Images were acquired using Zeiss LSM780 confocal laser scanning microscopes with ZEN software (Carl Zeiss, Jena, Germany) and Olympus FV3000 microscopes with FV31S-SW software (Olympus, Tokyo, Japan). Surface models of 3D reconstructed images were generated on Imaris ver.9 (Carl Zeiss).

**Identification and quantification of the hepatobiliary connections in HBTOs**. HBTOs were stained with phalloidin, anti-CK19, and anti-HNF4α antibodies, and then the phalloidin (+) luminal structures connecting CK19(+) ducts with HNF4α (+) hepatocyte clusters were counted. The same images were used to measure the boundaries between hepatocyte clusters and the biliary tissue, using Olympus cellSens software. For this analysis, HBTO culture in one well of the 24-well plate was repeated five times, independently ($n = 5$). The number of hepatobiliary connections was examined in more than eight different fields in each culture well. Dot plots were generated on GraphPad Prism ver. 5 (GraphPad, San Diego, CA).

**Examination of lineage plasticity of hepatocytes and cholangiocytes in HBTO**. tdTomato[+] SHs and wild-type cholangiocytes, or wild-type SHs and tdTomato[+] cholangiocyte were used for inducing HBTO. Both Wt SH and tdTomato[+] cholangiocyte and tdTomato[+] SH and Wt cholangiocyte culture was repeated three times, independently. After a month of co-culture, samples were stained with anti-HNF4α and phalloidin or anti-CK19 and phalloidin. Images were taken for two different areas in each sample to count tdTomato[+] cells. For co-culture of tdTomato[+] SHs and wild-type cholangiocytes, CK19 was used to find tdTomato[+]CK19[+] cells, which suggest hepatocyte-to-cholangiocyte conversion proceeds in HBTO. For co-culture of wild type SH and tdTomato[+] cholangiocytes, HNF4α was used to find tdTomato[+]HNF4α[+] cells, which suggest cholangiocyte to hepatocyte conversion proceeds in HBTO. In total, 590 and 280 tdTomato[+] cells were examined to detect tdTomato[+]HNF4α[+] and tdTomato[+]CK19[+] cells, respectively.

**Uptake of chloromethyl fluorescein diacetate (CMFDA) and cholyl-lysine fluorescein (CLF)**. Three to four weeks after Col-MG overlay, the medium was replaced with a differentiation medium containing 1 µg/ml CMFDA (FUJIFILM Wako Pure Chemical Corporation, Osaka, Japan) or 1 µg/ml CLF (Corning). The wells were washed with the differentiation medium five times, after 10 min incubation with CMFDA and 30 min incubation with CLF. Images were taken with an Olympus fluorescence microscope. Three to four areas per well were selected, and the transport of CMFDA and CLF was examined by taking fluorescence and phase-contrast images at different time points. Experiments were repeated three times ($n = 3$). Representative images are shown in Fig. 3. For quantification of CLF transport, BD areas were cropped using Photoshop (Adobe Systems, San Jose, CA) and the fluorescence intensity in the area was quantified using Image J ver. 1.48[33]. The results are shown in Supplementary Fig. 9. In addition to taking time courses of CLF transport, uptake of CLF during 12 h was examined three times, independently. In those experiments, CLF was detected in BCs and in the BDs connecting to the BCs.

**Detection of bilirubin in the organoids**. Ten mg of bilirubin (Kanto Chemicals, Tokyo, Japan) was dissolved in 0.1 ml of DMSO. Then 200 µl of 0.1 M $Na_2CO_3$, 500 µl of FBS, 20 µl of 0.1 N HCl, and 180 µl of distilled water were added. The solution was diluted ten-fold with DMSO. The bilirubin solution was further diluted 100-fold with culture medium, to produce a 10 µg/ml bilirubin solution, which was then filtered using a pore size of 0.22 µm[34]. HBTOs were incubated with a medium containing 10 µg/ml bilirubin for five days. Hall's method[35] was used for the histochemical staining of bilirubin. The organoids were washed with PBS and fixed with 10% buffered neutral formalin for 5 min at room temperature, and then washed twice with distilled water. Fouchet's reagent, containing 22.5% trichloroacetic acid and 1% ferric chloride, was added, and the mixture was incubated at room temperature for 5–15 min until green biliverdin was detected in the luminal networks of the organoids. Bilirubin detection was repeated three times ($n = 3$) three or four weeks after Col-MG overlay. A representative image is shown in Fig. 3.

**ALB ELISA**. Sandwich ELISA using goat anti-mouse ALB (Bethyl Laboratories, Montgomery, TX) and HRP-conjugated anti-mouse ALB antibodies (Bethyl Laboratories) was performed to measure ALB in the culture medium. Signal was detected using o-Phenylenediamine (OPD) (Sigma-Aldrich) and measured on an 800TS absorbance reader (BioTek, Winooski, VT). The HBTOs were kept for two to three months, and the ALB concentration in the culture medium was measured using ELISA every two weeks after Col-MG overlay. The long-term culture in one well of 24 well plates for eight weeks after Col-MG overlay was repeated five times ($n = 5$). Two out of five cultures were extended for additional four weeks. As a control, SHs were plated onto collagen gel in 24-well plates at a density of $5 \times 10^4$ cells/well and overlaid with Col-MG. In addition to five culture wells, one more well was used to determine ALB concentration in the culture medium four weeks after Col-MG overlay ($n = 6$).

**CYP activity**. CYP activity was measured using Glo-CYP3A4-Assay and Glo-CYP1B1-Assay (Promega, Madison, WI). The latter can be used to detect both CYP1A1 and 1B1. However, since the liver expresses CYP1A1 but not CYP1B1, the substrate provided in Glo-CYP1B1-Assay is specifically metabolized by CYP1A1 in hepatocytes. The substrate was added to the culture medium at the concentration shown in the protocol and incubated for one and three hours to measure CYP3A4-like activity and CYP1A1 activity, respectively. Luminescence was measured on Infinite M1000 Pro (Tecan Group Ltd., Männedorf, Switzerland). CYP1A1 and CYP3A4-like activities shown in Fig. 4 were sequentially assessed in three wells ($n = 3$) of an HBTO culture. One culture with three wells was repeated for measuring CYP1A1, and two more were repeated to measure CYP3A4-like activity. As a control, SHs alone were cultured in the same sandwich culture condition as HBTOs, and the CYP activity was examined. The CYP activities shown in Fig. 4 were normalized to the genomic DNA of hepatocytes. To quantify the genomic DNA of hepatocytes in HBTOs, tdTomato SHs were co-cultured with wild-type cholangiocytes. Genomic DNA was extracted using Nucleospin tissue (Takara Bio Inc., Shiga, Japan) and used for quantitative PCR detecting sequence of tdTomato. The primers used for PCR are shown in Supplementary Table 4. A standard curve was generated using differently diluted genomic DNA of tdTomato MHs, whose concentrations were measured on a Nanodrop (Thermofisher Scientific). The genomic DNA content of each sample was calculated from the threshold cycle (Ct) of the tdTomato signal based on the following formula.

$$DNA\,(ng/\mu l) = 2E + 22Ct^{-14.62}$$

**Culture of MHs**. MHs were isolated from the mouse liver by two-step collagenase perfusion. They were plated in a 24-well plate coated with type I collagen at the density of $1 \times 10^5$ cells/well. The CYP3A4-like activity was measured at one and five days after plating. MH culture and the assay for CYP activity were repeated twice. The average values of three wells in a representative culture are shown in Supplementary Fig. 10.

**Uptake of low-density lipoprotein**. Four weeks after Col-MG overlay, HBTOs were incubated with the growth medium containing 2 µg/ml DiI-acetylated low-density lipoprotein (LDL) (Alpha Acer, Havehill, MA) for 1 h. Wells were washed with the growth medium twice, and images were taken with an Olympus fluorescence microscope. The uptake of LDL was examined twice ($n = 2$) and representative images are shown in Fig. 4.

**RNA sequence analysis**. SHs and MHs were isolated from three healthy adult mice. SHs were further purified as CD31[−]CD45[−]EpCAM[−]ICAM-1[+] cells by FACSAriaII[9] (BD biosciences, San Jose, CA). Total RNA was extracted using the RNeasy Mini kit (Qiagen). Poly(A) RNAs were purified and cDNA libraries were constructed using Ion Total RNA-Seq Kit v2 (Thermo Fisher Scientific). RNA sequencing was performed on an Ion Proton system (Thermo Fisher Scientific). RNA-Seq data were analyzed using CLC Genomics Workbench (Qiagen Bioinformatics) to identify the differentially expressed genes. The raw transcriptome data were trimmed with quality scores limit set to 0.01, and the ambiguous limit set to 2. Trimmed reads of cDNA libraries were then aligned to the *Mus musculus* genome downloaded from the NCBI database (mm10) with the following parameters:(1) the maximum number of allowed mismatches 2;(2) minimum length fraction was set at 0.8;(3) minimum similarity fraction was set at 0.8; and (4) the maximum number of hits per read was 10. Gene expression values were reported as RPKM (Reads Per Kilobase of exon per Million mapped reads) using the CLC transcriptomic analysis module. RNAseq analysis was performed on a set of SHs and MHs. Therefore, we confirmed the results by quantitative PCR.

**Quantitative PCR**. MHs and SHs were isolated as described above in "Cell Isolation." Total RNA was extracted using an RNeasy mini kit according to the manufacture's protocol (Qiagen, Hilden, Germany). RNA was quantified on a Nanodrop (Thermo Fisher Scientific, Waltham, MA) and 50 ng of total RNA was used to synthesize cDNA using a PrimeScript 1st strand cDNA synthesis kit (Takara Bio Inc.). Quantitative PCR was performed using an ABI PRISM 7500 (Thermo Fisher Scientific) with the primers listed in Supplementary Table 4. Taqman gene expression assays for *Cps1*, *Hnf4a*, and *Hprt* (Thermo Fisher Scientific) were used for quantifying the expression of *Cps1* and *Hnf4a*. Isolation of MHs and SHs from a mouse was repeated eight times, independently. The expression levels were first normalized against those of *Gapdh or Hprt1*. The expression values in SHs were divided by those in MHs isolated from the same mouse and then used for statistical analysis. Expressions of *Cdh1* and *Cldn2* were

examined using 8 sets of SHs and MHs ($n = 8$) (Fig. 5), whereas those of other genes were using 4 sets of cells ($n = 4$) (Supplementary Fig. 15).

For analyzing the expression of cholangiocyte markers in HBTOs, a gel containing HBTOs was transferred from one well of a 24 well plate to a 2 ml tube, added with 500 μl of lysis solution, and then homogenized. Cholangiocytes cultured in the same condition as HBTOs was used as the control. Cultures and quantitative PCR were repeated four times independently ($n = 4$).

**Cell isolation from HBTO**. HBTOs were generated in a 12 well plate. Gels containing HBTOs were transferred from three wells into a 15 ml tube. Two ml of Cell Recovery Solution (BD biosciences) was added and incubated for 30 min on ice. Gels were broken down by pipetting and added with 8 ml of culture medium. After centrifugation for 4 min at $150 \times g$, the pellet was resuspended in 2 ml PBS containing Liberase TM (Roche) and incubated at 37 °C for 15 min. Gels were dissolved by further pipetting and centrifuged for 4 min at $150 \times g$. Cells were incubated with APC-conjugated anti-EpCAM antibody. EpCAM(−) and (+) cells were isolated on a MoFlow (Beckman coulter, Brea, CA). Cell isolation was repeated twice and the isolated cells were used for quantitative PCR.

**Uptake of Rho123 to the biliary tissue in HBTO**. HBTOs cultured for four weeks after Col-MG overlay was further incubated with a medium containing 10 μM Rho123 (FUJIFILM Wako Pure Chemical Corporation) for 30 min with or without 10 μM verapamil (Sigma-Aldrich). Wells were washed with medium five times, and images were taken on a fluorescence microscope.

**Effect of Forskolin on the biliary tissue in HBTO**. HBTOs cultured for four weeks after Col-MG overlay was further incubated in the presence of 10 μM Forskolin (FUJIFILM Wako Pure Chemical Corporation). Images were taken at 0 and 1 h after Forskolin treatment. To estimate the lumen expansion, the luminal areas with bright color were extracted using ImageJ ver. 1.48 and quantified.

**Transmission electron microscopy (TEM)**. The cells were fixed with phosphate-buffered saline (PBS) containing 2.5% glutaraldehyde at 4 °C for 10 min. After washing in 0.1 M PBS, the sample was incubated in PBS containing 1% $OsO_4$ and 1.5% potassium ferrocyanide for two hours. The cells were dehydrated by incubating through a concentration gradient of ethanol increasing up to 100%. The dehydrated samples were immersed in propylene oxide and then embedded in Epon (FUJIFILM Wako Pure Chemical Corporation). TEM samples were prepared from two independent culture samples. The samples were examined by using a transmission electron microscope (H7650; Hitachi, Tokyo, Japan).

**Separation of ECAD(+) MHs**. MHs were incubated with PE-conjugated anti-ECAD antibody (BioLegend) and propidium iodide (PI) (Dojindo Laboratories, Kumamoto, Japan). After selecting live cells by gating PI (−) cells, ECAD(−), and (+) MHs were isolated using FACSAria II. Five days before cell isolation, cholangiocytes were plated on type I collagen gel. Isolated MHs were plated onto collagen gel containing cholangiocyte colonies. Cell isolation and co-culture using ECAD(−) or ECAD(+) MHs in one well of a 24-well plate was repeated three times ($n = 3$). The number of hepatobiliary connections was counted in three to four areas in each culture well, after staining with antibodies against HNF4α and CK19, phalloidin, and Hoechst 33342.

**Culture of human reprogramming hepatocytes**. Human reprogramming hepatocytes (hCLIPs) were generated from primary human hepatocytes, as previously reported[22]. For inducing hybrid HBTO, mouse cholangiocytes isolated from tdTomato mice were plated on type I collagen gel. On the same day, a frozen stock of hCLIP was thawed and the hCLIPs were kept on dishes coated with type I collagen. On day 5, hCLIPs were treated with trypsin-EDTA and, after centrifugation, the cells were plated onto collagen gel, where cholangiocytes formed colonies. Two days after the hCLIPs were plated, 10 ng/ml human OSM was added to the culture, followed by an overlay of collagen gel containing 40% MG. The co-culture was repeated four times ($n = 4$). Representative data are shown in Fig. 6. The number of hepatobiliary connections was counted in four to seven areas in each culture well, after staining with antibodies against HNF4α and CK19, phalloidin, and Hoechst 33342.

**Isolation and culture of pancreatic duct cells**. Pancreatic duct cells were isolated from the mouse adult pancreas according to a previous report, with some modification[36]. After two-step collagenase perfusion and removal of the liver, the pancreas was resected from the mouse. The pancreas was placed in ice-cold PBS, then transferred to a 6 cm tissue culture dish, and minced into pieces smaller than 1 mm³. Ten milliliters of DMEM/F12 medium was added and pancreatic tissue pieces were transferred to a 50 ml tube. Floating adipose tissue debris was eliminated with a pipette. The pancreatic tissue pieces were resuspended in 25 ml DMEM/F12 medium, transferred into a 30 ml beaker containing a magnetic stir bar, and added with 25 mg collagenase. The cell suspension was incubated in a hybridization oven set to 37 °C with stirring at low speed for about 20 min. The digestion was terminated by diluting the collagenase solution with ice-cold PBS

containing 0.9 mg/ml glucose and 47.6 μM $CaCl_2$. The cell suspension was transferred to a 50 ml tube and centrifuged at $300 \times g$ for 5 min at 4 °C. The pellet was resuspended in 1 ml of trypsin-EDTA with a pipette and incubated at room temperature for 5 min. The digestion was terminated by adding 2 ml of DMEM/DF12 containing 10% FBS and centrifuged at $300 \times g$ for 5 min at 4 °C. The pellet was resuspended in DMEM/DF12 containing 10% FBS and used for the isolation of EpCAM⁺ pancreatic duct cells by MACS. Pancreatic duct cells were cultured on a collagen gel in the same condition as cholangiocytes and used for co-culture with SHs. For FACS analysis, PI(−) live singlet cells were selected by gating and used to detect EpCAM⁺ECAD⁺ pancreatic duct cells. Organoids consisting of hepatocytes and pancreatic ducts were stained with anti-HNF4α, FITC-conjugated dolichos biflorus agglutinin (DBA) (Vector Laboratories, Burlingame, CA), a lectin expressed in pancreatic ducts[37].

**Data analysis of FACS**. FACS data were analyzed and density and histogram plots were generated on Karuza ver 1.1. (Beckman coulter).

**Statistical analysis**. Unpaired two-tailed Student's $t$-tests were performed for the data pertaining to CYP1A1 and CYP3A4 activity, ALB secretion, and counting hepatobiliary connections, using Microsoft Excel. Paired two-tailed Student's $t$-tests were used to analyze the quantitative PCR data. When the result of a $t$-test showed $P < 0.05$, we concluded that the difference between the two groups was statistically significant. The numbers of culture sample used for each analysis were as follows: $n = 5$ (the number of hepatobiliary connection), $n = 5$ (ALB secretion), $n = 3$ (CYP activities), $n = 3$ (qPCR for SHs and MHs), $n = 8$ (the number of hepatobiliary connection in HBTO with ECAD(−) and ECAD(+) hepatocytes). For technical replicates, duplicates of each sample were generated and the average values were used in statistical analyses for data of CYP activities and qPCR.

**Reporting summary**. Further information on research design is available in the Nature Research Reporting Summary linked to this article.

## Data availability

RNA sequence data have been deposited in the Gene Expression Omnibus database under the accession code: GSE166283. All other relevant data are available from the corresponding author. Source data are provided with this paper.

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

## Acknowledgements

We thank Ms. Yumiko Tsukamoto for technical assistance. We thank Dr. Taketomo Kido and Dr. Yasuhiro Nakano for technical assistance and helpful discussion. We also thank all the members of Mitaka and Miyajima Laboratories for technical assistance and helpful discussion. We thank Olympus for technical assistance in acquiring confocal images at TOBIC. This work was supported by the Ministry of Education, Culture, Sports, Science and Technology, Japan, Grants-in-Aid for Scientific Research (C) for N.T. (25460271, 16K08716, 20K05843), Grants-in-Aid for Scientific Research on Innovative Areas "Stem Cell Aging and Disease" for N.T. (17H05653), Grants-in-Aid for Scientific Research (B) for T.M. (18H02873), and Grants-in-Aid for Exploratory Research for T.M. (26670584, 17K19703). This work was also supported by a Research Grant from the Orange Fund for the Commemoration of Hokkaido Hepatitis B Lawsuits for N.T. This work is partly supported by the Japan Agency for Medical Research and Development (AMED) for N.T. (JP20gm6210029h0001), and Research Program on Hepatitis from AMED for T.O. (JP19fk0210048, JP20fk0210048, JP20bm040442h0002).

## Author contributions

N.T.: writing (original draft), conceptualization, investigation, methodology. N.I.: data discussion, writing (review). Y.S.: data curation, formal analysis. T.I.: writing (review and editing), resources. R.S.: writing (editing), resources. T.Y.: resources. T.K.: resources. N.N.: transmission electron microscopic analyses. T.T.: data curation, formal analysis. T.O.: resources. A.M.: data discussion, resources. T.M.: writing (review and editing), funding acquisition.

## Competing interests

The authors declare no competing interests.
