## [Peer Review File · Nature Communications]

Reviewers' Comments:

Reviewer #1:

Remarks to the Author:

GENERAL REMARKS

The authors purified cholangiocytes (biliary cells) and a sub-population of hepatocytes from mouse livers. The cholangiocytes were cultured first and the hepatocytes were added subsequently, followed by Matrigel overlay. Connections were then detected between hepatocyte canaliculi and the lumina that had formed between adjacent cholangiocytes. These connections enabled metabolites to flow from hepatocyte canaliculi to the biliary lumina. The authors provide some evidence that differentiation is maintained and that the metabolic properties of the cells are improved in the co-culture conditions, as compared to cultures containing only hepatocytes. The authors conclude that they have generated functional hepatobiliary organoids in which hepatocytes and cholangiocyte functionally interconnect.

The topic of the work is important in biology as it addresses the question of how two distinct cell types connect with each other. In hepatology research, understanding how hepatocyte-cholangiocyte connections are established is essential for the construction of liver tissue for regenerative therapy.

The main strength of the work is in the generation of a liver-like organoid where metabolites can flow from hepatocyte canaliculi to a biliary luminal network. Unfortunately, the work has a number of weaknesses. First, the biliary luminal structures in the organoids do not convincingly resemble *in vivo* bile ducts. Second, the maintenance of the metabolic and differentiation properties of the cells are not well described. Third, the materials and methods section is below standards; it does not allow the reader to reproduce the experiments nor to fully understand the characteristics of the purified cells. Fourth, the use of human cells would have had considerably more impact.

MAJOR REMARKS

1. The biliary luminal structures formed between adjacent cholangiocytes in the cultures differ significantly from those in normal liver. Their morphology is reminiscent of hepatocyte canaliculi, not of bile ducts. To convince the reader that biliary luminal structures are biliary and not hepatocyte-like or hybrid hepato-biliary, the authors should stain their cultures to detect CEACAM1 (canaliculus marker) and Na⁺/H⁺ exchanger regulatory factor 1 (NHERF1; bile duct marker. Cfr Benhamouche-Trouillet et al. *Development* 2018). See also major comment 4 for differentiation-related remarks.

Along the same lines, the forskolin-induced dilation of biliary luminal structures is not convincing (Fig. S5). The main difference between the 0 and 1h time points is in the contrast between the two pictures. The reviewer suggest to homogenize the picture quality and quantify the luminal space before and after forskolin treatment.

2. The "small hepatocytes" are claimed to be more similar to zone 1 and zone 2 hepatocytes, than to zone 3 hepatocytes. This conclusion is based on the expression of Cyp's, membrane proteins (E-Cadherin, Claudin2) and Wnt target genes at the time of purification from mouse livers. Since the authors infer that similarity to zone1/2 hepatocytes facilitates the establishment of connections with the biliary network, the authors should show that the expression of zone1/2-specific genes by the "small hepatocytes" is maintained in culture.

3. The authors conclude that the "small hepatocytes" co-cultured with cholangiocytes form hepatobiliary organoids which express higher levels of Cyp's and Albumin than small hepatocytes cultured alone (Fig. 5). This is important as it suggests that hepatocyte maturation is improved

when hepatocytes establish connections with biliary cells. However, the authors' conclusions are based on RT-PCR data from RNA extracted from whole cultures, meaning that spatial information is lost. The authors should illustrate the expression of hepatocyte-specific enzymes by in situ hybridization or immunostaining to unambiguously demonstrate which cell type is performing which function in the co-culture.

4. Cholangiocyte differentiation is plastic and the medium contains, at some stage, dexamethasone and Oncostatin M, which are known to promote hepatocyte differentiation. Therefore, one may wonder whether the cultured cholangiocytes remain stably differentiated during the culture, or whether they acquire some hepatocyte metabolic properties, thereby contributing to the global metabolic functions of the organoids. In other terms, in parallel with the demonstration that only small hepatocytes exert the observed metabolic functions (comment 3), the authors should show how the expression of cholangiocyte differentiation markers evolves as a function of time (e.g. those measured by the authors, plus others commonly used such as Sox9, HNF1-beta, presence of primary cilia).

Of note, Fig. 3 suggests that the hepatocyte and cholangiocyte lineages are maintained as a function of time, but there is no unit of time mentioned in the figure.

5. The purity of the isolated "small hepatocytes" and cholangiocytes is not illustrated. Can the authors eliminate that some "small hepatocytes" connected with cholangiocytes are co-purified, and that such connected cells may prime the culture for further extension of the connections ?

6. How efficient is the organoid to establish hepato-biliary connections ? Fig. 6A measures the number of connections per millimeter of hepatocyte-cholangiocyte boundary, but the definition of a connection is not clear. For instance, in Fig. 3D, such a connection is illustrated and involves at least two hepatocytes and three or four cholangiocytes. One may expect that the number of cells involved varies throughout the culture, thereby affecting the functionality of the connections. How did the authors define a connection in their quantifications ?

7. Description of materials and methods is superficial and does not easily allow to reproduce the experiments. In particular it is not clear if cholangiocytes and "small hepatocytes" were purified according to the protocol described in the authors' previous work (Tanimizu et al. Stem Cells 2016). Differentiation medium is not defined either. Also, statistical analysis is insufficient, there is no clear mention of the number of replicates.

MINOR REMARKS

1. Fig. 1. Cholangiocytes are not easily identified in panel A; panel B is in part redundant with earlier published work (Fig. 5 in Tanimizu et al. Hepatology 2016;64:175-188). The reviewer suggests to present Fig. 1 as supplementary data.

2. There is considerable overlap between the conclusions of Figs. 2 and 3. Their description can be shortened and the two figures can be fused.

3. Reference 20 is not appropriate to illustrate zonation. There are excellent reviews on this topic, as well as spatial transcriptomic papers.

Reviewer #2:

Remarks to the Author:

Liver organoids have been developed in the last few years featuring hepatocytes or cholangiocytes. This work is the first to show an organoid with a functional connection between hepatocyte canaliculi and cholangiocyte bile ducts, an important component of liver function. This

new organoid system allows the study of the excretion of bile acids and waste products produced by hepatocytes. To make the organoids the authors use mouse small hepatocytes and cholangiocytes in a sandwich culture system. Under optimized conditions, small hepatocytes formed canaliculi that were able to connect to tubular structures formed by cholangiocytes, and the culture was maintained for 1 month. Functionality of these hepatobiliary connections was demonstrated by adding substrates that would be taken up by hepatocytes and then shown to accumulate in the bile ducts. Additionally, the authors found that some hepatocyte functions, such as albumin excretion and activity of some CYPs was improved in the organoids compared to hepatocytes cultured alone. Small hepatocytes form connections with cholangiocyte ducts in the organoids better than an unrefined population of mature hepatocytes, and small hepatocytes are enriched in zone 1 and 2 markers, suggesting that hepatocytes closer to the portal tracts more readily form connections with cholangiocytes, possibly because of higher E-Cadherin expression.

Overall, the work is well executed and demonstrates an important step towards building functional liver organoids. More data to back up some of the observations made in the text, more quantification of what is shown in the images, and more succinctly organized figures would make this a stronger manuscript. Specific comments are listed below.

Major comments:

- 1) Could the data be presented more succinctly so that it is easier for the reader to follow. For example, Fig. 1-3 and 6A all illustrate the rationale behind and formation of the organoids with hepatobiliary connections. The data potentially could be reduced to one or two figures highlighting the most important points with supporting and additional evidence shown in supplemental figures. The same could be said of Fig. 6B-E and Fig. 7.
- 2) Are Actin and ZO-1 definitive markers of bile canaliculi and luminal surface of bile ducts? Could stains for additional canaliculi markers such as DPPiV/CD26 or CD10 be shown? Could the same immunostains in liver tissues (Fig. 1) be shown in the organoids? An ideal combination would be OPN staining the luminal surface of cholangiocytes showing connection to a hepatocyte canaliculus (i.e. CD26+).
- 3) In Fig. 3, lack of interconversion between hepatocyte and cholangiocyte fate should be demonstrated by quantification.
- 4) In Fig. 4, include additional intermediate timepoints and quantification of dye intensity in the hepatocyte canaliculi vs. cholangiocyte ducts over time. Would be ideal if a movie could be made showing transport of one of the substrates from a canaliculus to a duct. Is it possible for some of the substrates to leak from hepatocyte canaliculi that are not connected to cholangiocyte ducts into the medium and then be absorbed and concentrated in cholangiocyte ducts?
- 5) How is cell viability and proliferation in the organoids over the course of the culture?
- 6) It would be informative to specify in the text and methods how mature hepatocytes and cholangiocytes alone were cultured. Were they also cultured under the same sandwich conditions with same media as the HBTO organoids?
- 7) Are the HBTO organoids changing over time as shown in Fig. 2? At what timepoint would they be considered a fully formed organoid and how long can that state be maintained? At what point of the culture was CMFDA, CLF, and bilirubin transport tested?
- 8) Is Table S1 missing?
- 9) From the main text: "During the postnatal period, the hepatobiliary luminal network becomes mature, as BC further extends and becomes the continuous luminal network"
 - a. The statement doesn't seem to be illustrated by the data shown
- 10) From the main text: "These data strongly suggest that SHs are a subfraction of hepatocytes in ZONE1 and ZONE2, which are more accessible to cholangiocytes within the liver tissue than those in ZONE3" and "Given that EpCAM cholangiocytes are positive for ECAD, these data are consistent with that SHs generate the functional connection with cholangiocytes more efficiently than MHs"
 - a. These statements can be verified. It should be possible to isolate hepatocytes from the 3 zones based on surface marker expression and compare how readily hepatocytes from the different zones form hepatobiliary junctions.
 - b. To test role of ECAD in hepatobiliary junction formation it should be possible to overexpress

ECAD in ECAD⁻ hepatocytes and knockdown ECAD in ECAD⁺ hepatocytes and determine efficiency of hepatobiliary junction formation.

11) Why not demonstrate HTBO organoids with primary human hepatocytes and cholangiocytes?

12) For discussion, can the organoids be developed into transplantable tissues?

Minor comments:

1. In the introduction, define small hepatocytes (SH) on their first mention, cite the previous work describing SH, and briefly summarize their unique characteristics. Should make clear that these are hepatocytes isolated from adult mice.

2. Arrows and arrowheads in the figures should be explained in the figure legends not main text.

3. Fig 1A: what is the white-appearing staining in the left "1" panel in the bile ducts?

4. Fig. 3: indicate timing in the schematics, what timepoint are the images in this figure from?

5. Fig. 3B: would be nice to have some cross sections as shown in Fig. 2E; the tdTomato makes it easier to see where the cholangiocytes are.

6. Fig 6C: images are too small, consider highlighting a few genes of interest in the main figure and showing the full graphs enlarged as a supplemental figure.

7. Fig 6E: use official gene name for glutamine synthetase (Glu1), which is used in 6D.

8. Fig. S1: include a nuclei stain, it is hard to tell where the cells are in B. Also, missing scale bars in these images.

9. Fig. S2: "tdTomato"? Use the conventional nomenclature for fluorophores

10. Fig. S4: were MH similarly cultured as HBTO?

11. Fig. S4: the red bars in the graphs are confusing; do they represent the activity levels in HTBO at days 1 and 5 of culture as well? It is better to show the HTBO data as a bar side-by-side to MH.

12. Fig S5C: can the biliary duct expansion with Forskolin be measured, quantified and shown in a graph?

13. Include more details in the methods section

a. "differentiation medium" is not clearly defined.

b. How often was media changed for the 1-month culture after the Matrigel overlay?

c. Size of wells can be important for organoid formation – what kind of culturing plate was used?

d. What is "collagen gel containing 20% Matrigel", is this a mixture of 80% 4mg/ml collagen + 20% Matrigel?

Reviewer #3:

Remarks to the Author:

In this study, the authors report the establishment of 'functional liver tissue' by coculture of mouse and hepatocyte progenitors and cholangiocytes.

This manuscript addresses a important need related to the development of a validated in vitro model for use in a range of studies, including disease modelling and drug screening. However, there are a number of major issues with the manuscript that make it difficult to draw definitive conclusions from the study.

Methods are not submitted as part of the paper, and no information is provided on the number of biological and technical replicates on the data presented. Much of the images are small and of low resolution, and it is impossible to know how representative the data are. Moreover, these images are all qualitative and no attempt is made to quantify or statistically compare the differences reported.

Some of the key experiments lack appropriate negative controls. For example, it would be necessary to show that other cell types (e.g., mammary epithelial cells) do not associated with hepatocytes in a similar manner as cholangiocytes (or conversely with hepatocytes).

The functional analysis of the hepatocytes should include enzymatic function (ALP). Phenotypic

analysis of the cells should include protein expression though IF in addition to qPCR.

The abstract states that this system is able to recapitulate 'drug metabolism in the liver', although no direct evidence is presented to support this claim.

Fig S1: lacks scale. Distinction between BCs and tubular networks is not convincing. Higher resolution images, ideally with 3D reconstruction are needed.

Fig 1. Generally low resolution and small; difficult to confirm the detail. Are these representative images? How many repeats from different animals were performed?

Fig 2. The BCs are difficult to see. How representative is the single white arrow in fig 2C5? How was the BC length measured in Fig2D? In general, the BCs are difficult to see and do not convincingly demonstrate the assertions.

Figure 4. Data before 10 minutes to show lack of fluorescein amongst cholangiocytes prior to 10 minutes; CLF administered to cholangiocytes alone as control.

Fig 6A: Difference between SH and MH data is not clear. 6B-D: control cells for MH and SH cells

Fig 7C: Should be shown on the same plot.

Reviewer #4:

Remarks to the Author:

This manuscript investigate new methodologies for the co-culture of bile ducts and hepatocytes as three-dimensional organoids. The approach itself is quite interesting and novel. I strongly recommend the authors to check the grammar/full sense of the manuscript with a native speaker. However, the manuscript is extremely poor, material and methods section insufficient and the results are unclear and repetitive.

Materials and Methods section is extremely poor, and I am quite concern that the descriptions provided in the text are completely different from what is presented in the figures.

Results section suffers insufficient detail regarding the description of the figures and the experimental results consistently failed to show proof for the manuscript's statements.

Results are over-interpreted, and the discussion does not match the conclusions of the study.

* I would not consider canaliculi as epithelial tissue as stated in the first paragraph. One thing is the bile duct, an anatomical tube made of biliary epithelial cells (cholangiocytes) and another are bile canaliculi, which are just the grooves on the lateral faces of hepatocytes. I think this is a fundamental histological mistake.

* The introduction has plenty of little typos. E.g. "Since hepatocyte clusters are functionally connected to the biliary tubules, we call this organoid AS hepatobiliary tubular organoid (HBTO) (AN hepatobiliary)"

* Lack of references for the co-cultivation of different cell populations to form organoids. Why don't they use references 8-13 here as well?

* When the authors mention that 'it can be assumed', is this assumption their hypothesis?

The methods are extremely unclear and lack entire sections.

* Authors need to include statistics section

* Please, include the type of gene expression profile analysis including bioinformatic data.

- * Please, include section stating the gene analysis performed in Fig 5. Include primers and RNA isolation, etc.
- * Software used for analysis of the images is stated in the text but not in the methods section
- * Please, include secondary antibodies used as well in the Supplementary table
- * Please, include the number of technical and biological replicas per experiment

RESULTS

There is overinterpretation of the data, repetitive experiments, and lack of clarity in the text. Text and figures are not coherent and are plagued with mistakes. As an example, the text states the assessment of Tdo2, Cyp3a11, and Cyp1a in panel B in Fig 6... where it cannot be found.

Sup Fig 1.

Bright field image of hepatocytes and bile ducts as single cell populations is not enough to prove their phenotype. Appropriate staining for the hepatocytes (HNF4a, Cyp2D6 etc) and cholangiocytes (EpCAM, Krt19 etc) should be performed. Bile canaliculi should be stained to be properly identified (DPPIV/CD26, CD25 etc).

If authors claim that they are similar to mature hepatocytes they should show images or mature hepatocytes with a similar phenotypic analysis (HNF4a, Cyp2D6, etc)

Are the figures the same magnification? Scale bar should be included

Fig 1.

A. Right panel (2) lacks clarity. The relevant area is almost hidden or out of the image.

B. The text states that this is developmental stage between 17 and 18 days. In the figure should be stated as 17.5 (or at least include it in the legend).

This result has already been stated in the literature and adds nothing new.

Fig 2.

A. What does the Panel B-1, Panel C-3 etc mean in the timeline? I cannot see to which part of the figure does this refer.

B. Please include all images in panel B at same bright contrast. Again, bright field is not enough to show that the two populations interact. Specially when the two populations cannot be distinguished. Fig Sup 2 is much clearer and should be included as part of Fig 2 (please, include scale bars in Fig Sup 2).

C. Why are the authors using Krt19 if they used OPN in the previous figure? Same goes for ZO-1 used before and Actin used here. Manuscript lacks consistency in the use of antibodies.

I recommend changing the color for Krt19 as it can barely be seen in the merge.

Is the staining phalloidin? Legend says phalloidin, image Actin? Please, reach a consensus

D. The sinus architecture is not a straight line. I do not think the method used to assess the extension of the BC is correct.

E. How many connections can they observe? Can they quantify this over the course of time to prove that there are more connections at 4 weeks as stated in the text?

Fig 3.

The text states that hepatocytes and cholangiocytes are plastic and can de-differentiate or transdifferentiate into cells forming HBTO. Included for this assumption is the reference from Raven et al., 2017. This citation is wrong. Please re-check the conclusions in the aforementioned manuscript. Also, none of these cells transform into HBTO (the new termed coined by the authors for their culture).

Bile canaliculi should be stained to be properly identified (DPPIV/CD26, CD25 etc).

Figures C and D could go into supplementary data as they are showing the same results with slightly different models. Actually, C and D look much clearer than A, B.

Fig S3.

Right panel is extremely blurry, and does not show what intended. Figure lacks scale bars.

Fig 4.

A. Why did the authors not make use of the TdTom model here to distinguish the cholangiocytes? Again, a bright field image is not enough to distinguish the two populations.

B. Same applies to panel B. Please consistently include the scale bars in the same images in all panels.

Fig 5.

A. Gene analysis lacks statistical analysis. Please, include the number of biological replicas per experiment. The text mentions that they express markers of metabolic activity, but compared to what? At what level? How is this experiment performed? Is all the co-culture lysed to obtain RNA? How can they infer then that the results purely belong to one population and not both?

Sup Fig 4 show a line for activity in comparison with other cell cultures. This is not the correct method to address this issue.

Sup Fig 5.

Panel A lacks statistics.

Fig 6.

A. Why do the authors compare now with MH when they have proved already that SH is a better method? Panels A and B should go into supplementary data as it adds nothing new.

B. The text mention the assessment of Tdo2, Cyp3a11, and Cyp1a in panel B. Where is this? Do they do the analysis using the co-culture of SH+BEC? Or the SH alone? Same for MH+BEC, or MH alone? The results are going to significantly differ depending on this.

This section concludes in the text with: 'Given that EpCAM+ cholangiocytes are positive for ECAD, these data are consistent with that SHs generate the functional connection with cholangiocytes more efficiently than MH' which has nothing to do with the experiment performed.

Responses to Reviewers' comments:

Reviewer #1 (Remarks to the Author):

We appreciate the valuable comments and suggestions from the reviewer. Our answers to the reviewer's comments are below.

MAJOR REMARKS

1. The biliary luminal structures formed between adjacent cholangiocytes in the cultures differ significantly from those in normal liver. Their morphology is reminiscent of hepatocyte canaliculi, not of bile ducts. To convince the reader that biliary luminal structures are biliary and not hepatocyte-like or hybrid hepato-biliary, the authors should stain their cultures to detect CEACAM1 (canaliculus marker) and Na⁺/H⁺ exchanger regulatory factor 1 (NHERF1; bile duct marker. Cfr Benhamouche-Trouillet et al. Development 2018). See also major comment 4 for differentiation-related remarks.

Along the same lines, the forskolin-induced dilation of biliary luminal structures is not convincing (Fig. S5). The main difference between the 0 and 1h time points is in the contrast between the two pictures. The reviewer suggests to homogenize the picture quality and quantify the luminal space before and after forskolin treatment.

To distinguish between the biliary lumen and bile canaliculi, we stained HBTOs with antibodies against CEACAM and Ezrin. Anti-Ezrin antibody specifically marks the biliary lumen (**Fig. S4**), as reported in a previous publication (**Hatano et al. Hepatology 2015**). The biliary lumen is Ezrin(+)CEACAM(-), and the bile canaliculi are Ezrin(-)CEACAM(+). The data indicate that Ezrin(+)CEACAM(-) and Ezrin(-)CEACAM(+) apical membranes surround the lumen at the junctions in HBTOs (**Fig. 2A**).

In accordance with the reviewer's suggestion, the lumen of the biliary structures was quantified using ImageJ, and it was clear that the lumen is expanded by forskolin treatment (**Fig. S10C**)

2. The "small hepatocytes" are claimed to be more similar to zone 1 and zone 2 hepatocytes, than to zone 3 hepatocytes. This conclusion is based on the expression of Cyps, membrane proteins (E-Cadherin, Claudin2) and Wnt target genes at the time of

purification from mouse livers. Since the authors infer that similarity to zone1/2 hepatocytes facilitates the establishment of connections with the biliary network, the authors should show that the expression of zone1/2-specific genes by the "small hepatocytes" is maintained in culture.

As the reviewer pointed out, the question of whether SHs maintain their original characteristics as ZONE1 and 2 hepatocytes is important. Hepatocytes in HBTOs show relatively low expression of *Cyp2e1* and *Cyp3a11* (**Fig. S8C**), both of which are strongly expressed in ZONE3 hepatocytes. This finding suggests that SHs somehow maintain their original characteristics in HBTOs. However, importantly, hepatocytes in HBTOs express CYP3A at a sufficient level to address CYP3A4-like activity, as shown in **Figures 4** and **S7**.

3. The authors conclude that the "small hepatocytes" co-cultured with cholangiocytes form hepatobiliary organoids which express higher levels of Cyp's and Albumin than small hepatocytes cultured alone (Fig. 5). This is important as it suggests that hepatocyte maturation is improved when hepatocytes establish connections with biliary cells. However, the authors' conclusions are based on RT-PCR data from RNA extracted from whole cultures, meaning that spatial information is lost. The authors should illustrate the expression of hepatocyte-specific enzymes by in situ hybridization or immunostaining to unambiguously demonstrate which cell type is performing which function in the co-culture.

In order to demonstrate that *Cyps* are expressed in hepatocytes but not in cholangiocytes in HBTOs, we separated the hepatocytes and cholangiocytes derived from HBTOs using FACS, and examined the expression of *Cyps* using qPCR. The data show that *Cyp1a2*, *2e1*, *3a11*, and *7a1* are exclusively expressed in hepatocytes (**Fig. S8C**). In addition, we showed that ALB is expressed in hepatocytes but not in cholangiocytes (**Fig. S9**). We agree with the reviewer's suggestion that it is better to show specific expression of CYPs in hepatocytes. Unfortunately, we have not been able to find effective antibodies for detecting CYPs in cultured hepatocytes. However, the qPCR data demonstrate that the CYP activity of HBTOs can be attributed exclusively to hepatocytes, and not to cholangiocytes.

4. Cholangiocyte differentiation is plastic and the medium contains, at some stage, dexamethasone and Oncostatin M, which are known to promote hepatocyte differentiation. Therefore, one may wonder whether the cultured cholangiocytes remain stably differentiated during the culture, or whether they acquire some hepatocyte metabolic properties, thereby contributing to the global metabolic functions of the organoids. In

other terms, in parallel with the demonstration that only small hepatocytes exert the observed metabolic functions (comment 3), the authors should show how the expression of cholangiocyte differentiation markers evolves as a function of time (e.g. those measured by the authors, plus others commonly used such as Sox9, HNF1-beta, presence of primary cilia).

Of note, Fig. 3 suggests that the hepatocyte and cholangiocyte lineages are maintained as a function of time, but there is no unit of time mentioned in the figure.

It is important to confirm that hepatocytes and cholangiocytes maintain their lineages in culture. We demonstrated that cholangiocytes barely differentiate into hepatocytes *in vitro*, even in the presence of dexamethasone and oncostatin M (Tanimizu et al. J Cell Sci 2013). In the work described in this revised manuscript, we stained HBTO with cholangiocyte markers, including SOX9, OPN, and EZN. These three markers were exclusively expressed in the biliary structures, but not in the hepatocytes, of HBTOs (Fig. 2). We could not identify any hepatocyte-to-cholangiocyte or cholangiocyte-to-hepatocyte conversions in HBTOs (Table S1). These data reflect the differentiation status of cholangiocytes one month after the overlay of collagen gel containing MG. We agree that it would be better to show the differentiation status of hepatocytes and cholangiocytes as a function of time, but our data definitely show that hepatocytes and cholangiocytes maintain their lineages during culture for at least a month, by which time we had assessed the metabolic functions and transport of hepatocyte metabolites.

Primary cilia were not clearly detected in biliary ducts in HBTOs under the electron microscope. As the reviewer pointed out, those biliary ducts may not be completely differentiated. In future, we may need to identify optimal conditions for further promoting their differentiation and maturation, both structurally and functionally. However, our aim in this work was to generate hepatobiliary connections *ex vivo*, and to provide a culture system recapitulating the transport of hepatocyte metabolites. Since bile acid and bilirubin absorbed by hepatocytes are transported to the biliary tissue via hepatobiliary connections in the HBTOs, we consider that the current differentiation status of hepatocytes, as well as cholangiocytes, in HBTOs is adequate to reproduce the transport of hepatocyte metabolites in liver tissue.

5. The purity of the isolated "small hepatocytes" and cholangiocytes is not illustrated. Can the authors eliminate that some "small hepatocytes" connected with cholangiocytes are co-purified, and that such connected cells may prime the culture for further extension of the connections ?

It is interesting that how the hepatobiliary connection is established in culture. Since SHs and cholangiocytes are enriched by centrifugation and MACS, respectively, the cell fractions were not 100% pure. However, as shown in **Figure S5** and **Table S1**, when cultures were started from Tomato(+)EpCAM(+) cells, they only formed biliary structures, and not hepatocytes, as evidenced by the observation that Tomato(+) hepatocytes did not emerge in HBTOs. We could not exclude the possibility that SHs connected to cholangiocytes contribute to making hepatobiliary junctions and eventually differentiated into biliary cells. However, as we described in the response to comment 4, cholangiocytes and SHs maintained their characteristics in culture, and barely changed their lineages (**Tanimizu et al. *J Cell Sci* 2013; *Sci Rep* 2017**). Therefore, we consider that neither contamination by hepatocytes in the cholangiocyte fraction nor cholangiocytes in SHs are majorly involved in hepatobiliary junctions.

6. How efficient is the organoid to establish hepato-biliary connections ? Fig. 6A measures the number of connections per millimeter of hepatocyte-cholangiocyte boundary, but the definition of a connection is not clear. For instance, in Fig. 3D, such a connection is illustrated and involves at least two hepatocytes and three or four cholangiocytes. One may expect that the number of cells involved varies throughout the culture, thereby affecting the functionality of the connections. How did the authors define a connection in their quantifications ?

The hepatobiliary connections were identified using structural analysis. HBTOs were stained with phalloidin, anti-CK19, and anti-HNF4 α antibodies, and then the phalloidin (+) luminal structures connecting CK19(+) ducts with HNF4 α (+) hepatocyte clusters were identified and counted. This method is described in the Materials and Methods section (**page 27**). As mentioned on page 7 of the manuscript, we found 3.4 ± 0.4 connections / 1 mm of the boundary between hepatocytes and cholangiocytes. Three to four different areas in four independent cultures were examined using this method. As the reviewer pointed out, the functionality of the connections may vary, as it is not clear whether this functionality depends on the number of cells involved in the connection.

7. Description of materials and methods is superficial and does not easily allow to reproduce the experiments. In particular it is not clear if cholangiocytes and "small hepatocytes" were purified according to the protocol described in the authors' previous work (Tanimizu et al. *Stem Cells* 2016). Differentiation medium is not defined either. Also, statistical analysis is insufficient, there is no clear mention of the number of replicates.

We apologize for not providing enough information in the Materials and Methods section. We have added the necessary information and rewritten the Materials and Methods, including the information that cholangiocytes and small hepatocytes were purified according to the protocols in J Cell Sci 2013 and Stem Cells 2016, respectively.

MINOR REMARKS

1. Fig. 1. Cholangiocytes are not easily identified in panel A; panel B is in part redundant with earlier published work (Fig. 5 in Tanimizu et al. Hepatology 2016;64:175-188). The reviewer suggests to present Fig. 1 as supplementary data.

We decided to provide the data in **Figure 1** as part of the Supplementary Material, and included it in **Figure S12**.

2. There is considerable overlap between the conclusions of Figs. 2 and 3. Their description can be shortened and the two figures can be fused.

We used the data in **Figure 3** in the new **Figure 2** and **Figure S5** to avoid overlap.

3. Reference 20 is not appropriate to illustrate zonation. There are excellent reviews on this topic, as well as spatial transcriptomic papers.

We have replaced reference 20 with a recent review, as follows:

Shani Ben-Moshe and Shalev Itzkovit. Spatial heterogeneity in the mammalian liver. *Nat. Rev. Gastroenterol Hepatol.* 2019, 16, 395

Reviewer #2 (Remarks to the Author):

Major comments:

1) Could the data be presented more succinctly so that it is easier for the reader to follow. For example, Fig. 1-3 and 6A all illustrate the rationale behind and formation of the organoids with hepatobiliary connections. The data potentially could be reduced to one or two figures highlighting the most important points with supporting and additional

evidence shown in supplemental figures. The same could be said of Fig. 6B-E and Fig. 7.

We modified **Figure 2** and included it in the new **Figure 1**, in which we demonstrate the luminal continuity between the bile canaliculi of hepatocytes and biliary ducts in culture. The data in **Figure 3** are used in the new **Figures 2** and **S5**, and **Table S1**. We also combined **Figures 6** and **7** and provide the result as the new **Figure 5**. In addition, we moved **Figure 1** to the Supplementary Materials and used it as part of **Figure S12**.

2) Are Actin and ZO-1 definitive markers of bile canaliculi and luminal surface of bile ducts? Could stains for additional canaliculi markers such as DPPIV/CD26 or CD10 be shown? Could the same immunostains in liver tissues (Fig. 1) be shown in the organoids? An ideal combination would be OPN staining the luminal surface of cholangiocytes showing connection to a hepatocyte canaliculus (i.e. CD26+).

As the reviewer pointed out, neither Actin nor ZO1 define bile canaliculi or the luminal surface of bile ducts. To distinguish between biliary lumen and bile canaliculi, we stained HBTOs with CEACAM and Ezrin. Anti-Ezrin antibody specifically marks the biliary lumen (**Fig. S4**) (**Hatano et al. Hepatology 2015**). As shown in **Figure 2**, the biliary lumen are Ezrin(+)/CEACAM(-) and the bile canaliculi are Ezrin(-)/CEACAM(+). We also show the OPN expression that was specifically detected in the biliary structures in HBTOs (**Fig. S3**).

3) In Fig. 3, lack of interconversion between hepatocyte and cholangiocyte fate should be demonstrated by quantification.

We examined the possibility of interconversion between hepatocytes and cholangiocytes in HBTOs derived from Tomato(+) cholangiocytes or Tomato(+) SHs (**Table S1**). We could not detect Tomato (+) cholangiocytes in a culture of wild type cholangiocytes and Tomato(+) SH, and *vice versa*. The data demonstrate that there was almost no interconversion between hepatocytes and cholangiocytes in HBTOs.

4) In Fig. 4, include additional intermediate time points and quantification of dye intensity in the hepatocyte canaliculi vs. cholangiocyte ducts over time. Would be ideal if a movie could be made showing transport of one of the substrates from a canaliculus to a duct. Is it possible for some of the substrates to leak from hepatocyte canaliculi that are not connected to cholangiocyte ducts into the medium and then be absorbed and concentrated in cholangiocyte ducts?

In cholangiocyte cultures, neither CMFDA nor CLF was incorporated into the luminal space (**Fig. S6**). The bile ducts accommodating CMFDA and CLF were always near the connections. These data support our conclusion that CMFDA and CLF taken up by hepatocytes are secreted to the bile canaliculi and then transported into the bile ducts in HBTOs. We could not take movies using our current experimental equipment.

5) How is cell viability and proliferation in the organoids over the course of the culture?

We do not believe there is very dynamic turnover of cells in HBTOs, but we assumed that hepatocytes and cholangiocytes may be slowly replaced by self-duplication of the remaining cells.

6) It would be informative to specify in the text and methods how mature hepatocytes and cholangiocytes alone were cultured. Were they also cultured under the same sandwich conditions with same media as the HBTO organoids?

In order to collect data about CYP activities and ALB secretion (**Figure 4**), we cultured SHs under the same sandwich conditions, with the same media, as the HBTOs. We describe the culture condition for SHs in the Materials and Methods section (**pages 26 and 27**). **Figure S7** shows MHs cultured in the same media as the HBTOs, but plated onto tissue culture dishes coated with type I collagen. The culture conditions for MHs are described in the Supplementary Methods.

7) Are the HBTO organoids changing over time as shown in Fig. 2? At what time point would they be considered a fully formed organoid and how long can that state be maintained? At what point of the culture was CMFDA, CLF, and bilirubin transport tested?

Functional assays were performed between three and four weeks after the overlay of collagen gel containing MG. As judged from the ALB secretion data, the organoids maintained their differentiation status for between 3 and 10 weeks. We also measured CYP3A4-like activity (**Fig. S7**), showing that HBTOs could maintain this activity for between two and six weeks.

8) Is Table S1 missing?

We apologize for incorrectly numbering the Supplementary Tables. We now provide Tables **S1**

to **S4**, showing the quantitative data relating to hepatocyte-cholangiocyte interconversion, the list of primary antibodies, the list of secondary antibodies, and the list of PCR primers.

9) From the main text: “During the postnatal period, the hepatobiliary luminal network becomes mature, as BC further extends and becomes the continuous luminal network”

a. The statement doesn’t seem to be illustrated by the data shown

As the reviewer pointed out, the data are not sufficient to support our conclusion. Therefore, the figure has been transferred to **Figure S12**, which shows that some hepatobiliary connections were detected in the late fetal period.

10) From the main text: “These data strongly suggest that SHs are a subfraction of hepatocytes in ZONE1 and ZONE2, which are more accessible to cholangiocytes within the liver tissue than those in ZONE3” and “Given that EpCAM cholangiocytes are positive for ECAD, these data are consistent with that SHs generate the functional connection with cholangiocytes more efficiently than MHs”

a. These statements can be verified. It should be possible to isolate hepatocytes from the 3 zones based on surface marker expression and compare how readily hepatocytes from the different zones form hepatobiliary junctions.

b. To test role of ECAD in hepatobiliary junction formation it should be possible to overexpress ECAD in ECAD- hepatocytes and knockdown ECAD in ECAD+ hepatocytes and determine efficiency of hepatobiliary junction formation.

In this revised manuscript, we co-cultured cholangiocytes with ECAD(-) and (+) MHs sorted using FACS, and demonstrated that the connections were more efficiently generated by ECAD(+) cells than by ECAD(-) MHs (**Figs. 5E and F**). It is difficult to specifically knock out ECAD expression in SHs in co-culture. In future, we will develop a method for performing this experiment.

11) Why not demonstrate HTBO organoids with primary human hepatocytes and cholangiocytes?

As the reviewer pointed out, it is important to generate HBTOs containing human hepatocytes. In this revised manuscript, we demonstrate HBTOs with human CLiP (chemically induced liver progenitors). hCLiPs are derived from primary human hepatocytes, and have the ability to differentiate into functional hepatocytes (**Katsuda et al. Cell Stem Cell 2017; eLife 2019**). We

co-cultured hCLiP with mouse cholangiocytes to generate HBTOs. This result is shown in the new **Figure 6**.

12) For discussion, can the organoids be developed into transplantable tissues?

We discuss the possibility of transplantation of HBTOs in the Discussion (pages 19 and 20). To take advantage of the features of HBTOs containing a bile excretion system, it is necessary to explore a technique connecting the biliary tissue in the HBTOs to the bile ducts of recipients, to secure permanent bile drainage. With this methodology, we would expect HBTOs to be engrafted in the recipient liver and maintain their functionality.

Minor comments:

1. In the introduction, define small hepatocytes (SH) on their first mention, cite the previous work describing SH, and briefly summarize their unique characteristics. Should make clear that these are hepatocytes isolated from adult mice.

Thank you for your suggestion. We have defined small hepatocytes in the first paragraph of the Results, indicating their characteristics and that they were isolated from adult mice.

2. Arrows and arrowheads in the figures should be explained in the figure legends not main text.

We added explanations about the arrows and arrowheads in the figure legends.

3. Fig 1A: what is the white-appearing staining in the left "1" panel in the bile ducts?

At this magnification, bile ducts positive for OPN (green), ZO1 (red), and Hoechst 33342 (blue) appear white on staining. In the revised manuscript, we used these data in **Fig. S12**.

4. Fig. 3: indicate timing in the schematics, what timepoint are the images in this figure from?

Immunostaining was performed using co-culture four weeks after Col-MG overlay.

5. Fig. 3B: would be nice to have some cross sections as shown in Fig. 2E; the tdTomato makes it easier to see where the cholangiocytes are.

We added optical cross sections of HTBOs with tdTomato SHs in **Figure 2B-5**.

6. Fig 6C: images are too small, consider highlighting a few genes of interest in the main figure and showing the full graphs enlarged as a supplemental figure.

In this revised version, we measured the expression of hepatocyte markers in hepatocytes and cholangiocytes purified from HBTOs using qPCR, and provide the data in **Figure S8**.

7. Fig 6E: use official gene name for glutamine synthetase (Glu1), which is used in 6D.

We changed Gs to Glu1 in **Figure S11C**.

8. Fig. S1: include a nuclei stain, it is hard to tell where the cells are in B. Also, missing scale bars in these images.

We added images containing a nuclear stain in **Figure S1**.

9. Fig. S2: “tdTomato”? Use the conventional nomenclature for fluorophores

Tomato is a red fluorescent protein and tdTomato is tandem dimeric Tomato (**Chaner et al. *Nat. Biotech.* 2004**). tdTomato has been widely used for genetic lineage tracing (**Raven et al. *Nature* 2017**).

10. Fig. S4: were MH similarly cultured as HBTO?

MHs were cultured on tissue culture dishes coated with type I collagen and maintained overnight before assessing CYP activity. The culture conditions for MHs are described in the Supplementary Methods.

11. Fig. S4: the red bars in the graphs are confusing; do they represent the activity levels in HTBO at days 1 and 5 of culture as well? It is better to show the HTBO data as a bar side-by-side to MH.

In accordance with the reviewer’s suggestion, we show the HTBO data as bars side by side with MHs (**Fig. S7**).

12. Fig S5C: can the biliary duct expansion with Forskolin be measured, quantified and shown in a graph?

We added the quantitative data for the duct expansion in **Figure S10C**.

13. Include more details in the methods section

a. “differentiation medium” is not clearly defined.

b. How often was media changed for the 1-month culture after the Matrigel overlay?

c. Size of wells can be important for organoid formation – what kind of culturing plate was used?

d. What is “collagen gel containing 20% Matrigel”, is this a mixture of 80% 4mg/ml

collagen + 20% Matrigel?

We have totally rewritten the Materials and Methods section, and provided the details about all of the experimental procedures, including those addressing the reviewer's comments on experimental procedures.

Reviewer #3 (Remarks to the Author):

In this study, the authors report the establishment of 'functional liver tissue' by coculture of mouse and hepatocyte progenitors and cholangiocytes.

This manuscript addresses an important need related to the development of a validated in vitro model for use in a range of studies, including disease modelling and drug screening. However, there are a number of major issues with the manuscript that make it difficult to draw definitive conclusions from the study.

1. Methods are not submitted as part of the paper, and no information is provided on the number of biological and technical replicates on the data presented. Much of the images are small and of low resolution, and it is impossible to know how representative the data are. Moreover, these images are all qualitative and no attempt is made to quantify or statistically compare the differences reported.

We apologize that we did not include the Material and Methods in the main text. We added detailed technical information and included it in the text.

2. Some of the key experiments lack appropriate negative controls. For example, it would be necessary to show that other cell types (e.g., mammary epithelial cells) do not associated with hepatocytes in a similar manner as cholangiocytes (or conversely with hepatocytes).

It is interesting that cholangiocytes, but not other tubular cells, can make connections with hepatocytes. However, we consider that the generation of hepatobiliary connections is a novel and important step toward generating functional liver tissue.

4. The functional analysis of the hepatocytes should include enzymatic function (ALP). Phenotypic analysis of the cells should include protein expression though IF in addition to qPCR.

To demonstrate the functionality of the hepatocytes, we provided an image of the uptake of acetyl-LDL by hepatocytes in HBTOs (**Fig. 4D**). It is important to show protein expression as part of the phenotypic analysis. We showed that ALB is expressed in hepatocytes but not in cholangiocytes forming HBTOs (**Fig. S9**).

Unfortunately, we could not obtain data about the expression of CYPs in organoids. Instead, in addition to data about CYP activity, we separated the hepatocytes and cholangiocytes derived from HBTOs and examined *Cyps* in both groups. The results showed that *Cyps* are specifically expressed in hepatocytes (**Fig. S8A-C**).

5. The abstract states that this system is able to recapitulate 'drug metabolism in the liver', although no direct evidence is presented to support this claim.

As the reviewer pointed out, we did not obtain data about drug metabolism. We changed the sentence to “This hepatobiliary organoid enables us to reproduce the transport of hepatocyte metabolites in liver tissue”.

6. Fig S1: lacks scale. Distinction between BCs and tubular networks is not convincing. Higher resolution images, ideally with 3D reconstruction are needed.

Scale bars have been added to **Figure S1**. To distinguish between BCs and biliary tubules, we examined HBTOs using CEACAM, a marker specific for BCs, and Ezrin, a marker specific for BDs. The data are presented in **Figure 2**.

7. Fig 1. Generally low resolution and small; difficult to confirm the detail. Are these representative images? How many repeats from different animals were performed?

We apologize that the images were not large enough to recognize the hepatobiliary structures. We have now provided magnified images focusing on the hepatobiliary junctions (**Fig. 2**). We repeated the cultures more than 50 times, using more than 200 wells. In each experiment, we isolated cholangiocytes from two mice, and five days later isolated small hepatocytes from two mice.

8. Fig 2. The BCs are difficult to see. How representative is the single white arrow in fig 2C5? How was the BC length measured in Fig2D? In general, the BCs are difficult to see and do not convincingly demonstrate the assertions.

We combined **Figures 2** and **3** to demonstrate the generation of lumen connectivity between hepatocytes and cholangiocytes (**Fig.1**). To demonstrate the hepatobiliary connections, we have

provided magnified images in **Figure 2**, in which luminal structures in the BC can be recognized.

In our culture system, hepatocytes form monolayers. Therefore, it is possible to draw polygonal lines along the BC in hepatocyte clusters and quantify them. We carried out the quantification using Olympus cellSens software.

9. Figure 4. Data before 10 minutes to show lack of fluorescein among cholangiocytes prior to 10 minutes; CLF administered to cholangiocytes alone as control.

In cholangiocyte cultures, CLF was not incorporated into the luminal space at two hours (**Fig. S6B**), or even at 24 hours (data not shown). The bile ducts containing CLF were always near the connections. These data support our conclusion that CLF is taken up by hepatocytes, secreted to bile canaliculi, and then transported into bile ducts in HBTOs. Unfortunately, it is difficult to take pictures before 10 minutes. In future, we will try setting up a system to take movies of CLF transport in HBTOs.

10. Fig 6A: Difference between SH and MH data is not clear. 6B-D: control cells for MH and SH cells

We apologize that the image was too small to detect the difference between SH and MH. We have now provided magnified images in **Figure 5A**. As indicated by arrowheads, lumen continuity between the BC and the bile ducts was apparent in co-cultures of SH and cholangiocytes. Co-cultures of cholangiocytes and MHs had very little lumen connectivity.

11. Fig 7C: Should be shown on the same plot.

We have provided a histogram showing ECAD expression in SHs and MHs (**Fig. 5D**).

Reviewer #4 (Remarks to the Author):

This manuscript investigate new methodologies for the co-culture of bile ducts and hepatocytes as three-dimensional organoids. The approach itself is quite interesting and novel. I strongly recommend the authors to check the grammar/full sense of the manuscript with a native speaker.

However, the manuscript is extremely poor, material and methods section insufficient and the results are unclear and repetitive.

We appreciate that the reviewer evaluated our work as novel. We have asked an editorial service to correct the text, and the text is now written in standard academic English. We have carefully rewritten the manuscript to demonstrate to the reviewer that our results are reproducible and reliable.

1. Materials and Methods section is extremely poor, and I am quite concern that the descriptions provided in the text are completely different from what is presented in the figures. Results section suffers insufficient detail regarding the description of the figures and the experimental results consistently failed to show proof for the manuscript's statements. Results are over-interpreted, and the discussion does not match the conclusions of the study.

We have extensively modified the content of the manuscript, including the Materials and Methods, to ensure consistency between our interpretation and the results shown in the figures. We have interpreted the results carefully, to avoid over-interpretation and over-discussion.

2. I would not consider canaliculi as epithelial tissue as stated in the first paragraph. One thing is the bile duct, an anatomical tube made of biliary epithelial cells (cholangiocytes) and another are bile canaliculi, which are just the grooves on the lateral faces of hepatocytes. I think this is a fundamental histological mistake.

As epithelial tissue structures, the liver contains biliary tubules and hepatic cords. In this work, we connected the lumens of bile ducts with those of hepatic cords (the bile canaliculi). The apical luminal networks of the bile ducts were connected to the bile canaliculi of the hepatic cords in our culture system. We rewrote the text to incorporate this histological interpretation.

3. The introduction has plenty of little typos. E.g. ‘Since hepatocyte clusters are functionally connected to the biliary tubules, we call this organoid AS hepatobiliary tubular organoid (HBTO) ◊ (AN hepatobiliary)’

We appreciate the reviewer pointing out our mistakes. We have carefully checked the text and asked a native speaker belonging to an editorial service to edit the English.

4. Lack of references for the co-cultivation of different cell populations to form organoids. Why don’t they use references 8-13 here as well?

We established the current protocol based on methods for inducing the differentiation and morphogenesis of hepatocytes and cholangiocytes, as reported in references 8-13. In the Discussion we compare our protocol with previous reports of cultures containing different cell populations (references 23 and 24).

5. When the authors mention that ‘it can be assumed’, is this assumption their hypothesis?

We have carefully checked the text and made our hypotheses clear.

6. The methods are extremely unclear and lack entire sections.

*** Authors need to include statistics section**

*** Please, include the type of gene expression profile analysis including bioinformatic data.**

*** Please, include section stating the gene analysis performed in Fig 5. Include primers and RNA isolation, etc.**

*** Software used for analysis of the images is stated in the text but not in the methods section**

*** Please, include secondary antibodies used as well in the Supplementary table**

*** Please, include the number of technical and biological replicas per experiment**

We have completely rewritten the Materials and Methods section, and included all of the information that should be provided.

We list the secondary antibodies in **Table S3**.

RESULTS

7. There is overinterpretation of the data, repetitive experiments, and lack of clarity in the text. Text and figures are not coherent and are plagued with mistakes. As an example, the text states the assessment of Tdo2, Cyp3a11, and Cyp1a in panel B in Fig 6... where it

cannot be found.

Thank you for pointing out our mistakes. We modified **Figure 2** into the new **Figure 1**, in which we demonstrate the luminal continuity between bile canaliculi of hepatocytes and biliary ducts in culture. The data in **Figure 3** are used in new **Figures 2** and **S5**, and in **Table S1**. We also combined **Figures 6** and **7** and provided them as the new **Figure 5**. In addition, we moved **Figure 1** to the Supplementary Materials, and used the data as part of **Figure S12**. We consider that these modifications make our data more comprehensible. We further checked our description in the text to ensure that it correctly explains the results shown in the figures.

Sup Fig 1.

8. Bright field image of hepatocytes and bile ducts as single cell populations is not enough to prove their phenotype. Appropriate staining for the hepatocytes (HNF4a, Cyp2D6 etc) and cholangiocytes (EpCAM, Krt19 etc) should be performed. Bile canaliculi should be stained to be properly identified (DPPIV/CD26, CD25 etc). If authors claim that they are similar to mature hepatocytes they should show images or mature hepatocytes with a similar phenotypic analysis (HNF4a, CYP2D6, etc)

We added data arising from staining with HNF4 α and ALB in **Figure S1**. To distinguish hepatocytes and cholangiocytes in HBTOs, we stained organoids with hepatocyte and cholangiocyte markers. As hepatocyte markers, we used HNF4 α , CEACAM, and Radixin. As cholangiocyte markers, we used CK19, Ezrin, SOX9, and OPN. In those images, it is clearly apparent that hepatocytes and cholangiocytes form the hepatobiliary connections in HBTOs (**Figs. 1C and Fig. 2**).

9. Are the figures the same magnification? Scale bar should be included.

We added scale bars to all figures.

10. Fig 1.

A. Right panel (2) lacks clarity. The relevant area is almost hidden or out of the image. B. The text states that this is developmental stage between 17 and 18 days. In the figure should be stated as 17.5 (or at least include it in the legend). This result has already been stated in the literature and adds nothing new.

We modified the panels according to the reviewer's suggestion, and transferred this figure to the

Supplementary Material (Fig. S12).

11. Fig 2.

A. What does the Panel B-1, Panel C-3 etc mean in the timeline? I cannot see to which part of the figure does this refer.

B. Please include all images in panel B at same bright contrast. Again, bright field is not enough to show that the two populations interact. Especially when the two populations cannot be distinguished. Fig Sup 2 is much clearer and should be included as part of Fig 2 (please, include scale bars in Fig Sup 2).

We modified **Figure 2** to include bright field images, and used it as the new **Figure 1**, in which we demonstrate luminal continuity between the bile canaliculi of hepatocytes and biliary ducts in culture. The data in **Figure 3** are used in the new **Figures 2** and **S5** to further illustrate the connection between hepatocytes and cholangiocytes (**Fig. 2B**).

C. Why are the authors using Krt19 if they used OPN in the previous figure? Same goes for ZO-1 used before and Actin used here. Manuscript lacks consistency in the use of antibodies. I recommend changing the color for Krt19 as it can barely be seen in the merge. Is the staining phalloidin? Legend says phalloidin, image Actin? Please, reach a consensus

To avoid any confusion, we transferred **Figure 1** to **Figure S12** for the Discussion. To show the tissue structure in the organoids, we changed the colors showing the expression of CK19. In addition, we changed the annotation “Actin” to “phalloidin” in the figures.

D. The sinus architecture is not a straight line. I do not think the method used to assess the extension of the BC is correct.

Thank you for this comment. As the reviewer pointed out, the sinus architecture is not a straight line. We therefore drew polygonal lines along the boundary and quantified their lengths using Olympus cellSens software.

E. How many connections can they observe? Can they quantify this over the course of time to prove that there are more connections at 4 weeks as stated in the text?

As mentioned on page 7 of the text, we evaluated the efficiency of establishing hepatobiliary connections in the organoids by counting the lumens connecting hepatocyte clusters and biliary tubules. We found 3.9 ± 0.4 connections / 1 mm of the boundary between hepatocytes and

cholangiocytes (**page 8**). We explain this method in the Materials and Methods section (**page 27**).

We did not evaluate changes in the number of connections over time. The hepatobiliary connections are mostly formed at an early time point, and subsequently the bile canaliculi becomes the connective network, resulting in a mature hepatobiliary network.

11. Fig 3.

The text states that hepatocytes and cholangiocytes are plastic and can de-differentiate or transdifferentiate into cells forming HBTO. Included for this assumption is the reference from Raven et al., 2017. This citation is wrong. Please re-check the conclusions in the aforementioned manuscript. Also, none of these cells transform into HBTO (the new term coined by the authors for their culture).

Bile canaliculi should be stained to be properly identified (DPPIV/CD26, CD25 etc).

Figures C and D could go into supplementary data as they are showing the same results with slightly different models. Actually, C and D look much clearer than A, B.

We apologize for causing confusion. We actually intended to show there was no conversion between hepatocytes and cholangiocytes in the culture. We have added quantitative data to this effect in **Table S1**. We cited a recent review about plasticity instead of the reference from Raven et al. 2017.

As mentioned above, we simplified **Figure 2** and provided it as the new **Figure 1**. Panels C and D of **Figure 3** are now **Figure 2B**. In addition, we transferred panels A and B to the Supplementary Data (**Fig. S5**).

As described in the responses to points 8 and 11, we stained HBTOs with hepatocyte-specific CEACAM and radixin to mark the canalicular membranes, and OPN and Ezrin to mark the lumen of bile ducts, to identify the nature of the hepatobiliary connections (**Figs. 2 and S3**).

12. Fig S3. Right panel is extremely blurry, and does not show what intended. Figure lacks scale bars.

We changed the image showing that bile ducts do not take up FDA, and added a new image demonstrating that hepatocytes, but not cholangiocytes, take up CLF. The images are in **Figure S6**.

13. Fig 4. A. Why did the authors not make use of the TdTom model here to distinguish the cholangiocytes? Again, a bright field image is not enough to distinguish the two populations. B. Same applies to panel B. Please consistently include the scale bars in the same images in all panels.

Thank you for your comments. We recognize that CLF uptake may be clearer using tdTomato cells. In future work, we will use tdTomato cells for HBTO culture to examine the transport of CMFDA and CLF.

14. Fig 5.

A. Gene analysis lacks statistical analysis. Please, include the number of biological replicas per experiment. The text mentions that they express markers of metabolic activity, but compared to what? At what level? How is this experiment performed? Is all the co-culture lysed to obtain RNA? How can they infer then that the results purely belong to one population and not both?

Sup Fig 4 show a line for activity in comparison with other cell cultures. This is not the correct method to address this issue.

We cultured SHs or HBTOs (SHs with cholangiocytes) under the same conditions, and measured the CYPs (**Fig. 4**). We describe the culture conditions for SHs in the Materials and Methods section (**pages 26 and 27**). We also compared the CYP3A4-like activity of HBTOs with that of primary MH cultures in the same graph (**Fig S7**). HBTOs maintain their CYP activity, whereas primary MHs initially have strong CYP activity, but rapidly lose the activity.

In the revised manuscript we separated hepatocytes and cholangiocytes derived from HBTOs and demonstrated that *Cyps* are specifically expressed in hepatocytes (**Fig. S8**).

15. Sup Fig 5.

Panel A lacks statistics.

We did not find statistically significant differences, suggesting that the cholangiocytes in HBTOs express cholangiocyte markers at levels similar to those of cholangiocytes cultured alone.

16. Fig 6.

A. Why do the authors compare now with MH when they have proved already that SH is a better method? Panels A and B should go into supplementary data as it adds nothing new.

B. The text mention the assessment of Tdo2, Cyp3a11, and Cyp1a in panel B. Where is this?

Do they do the analysis using the co-culture of SH+BEC? Or the SH alone? Same for MH+BEC, or MH alone? The results are going to significantly differ depending on this.

This section concludes in the text with: “Given that EpCAM+ cholangiocytes are positive for ECAD, these data are consistent with that SHs generate the functional connection with cholangiocytes more efficiently than MH” which has nothing to do with the experiment performed.

As the reviewer pointed out, we have demonstrated that SHs are a better source of cells for HBTOs than MHs. However, in addition to this observation, we identified that SHs and MHs differentially express As shown in new **Figures 5E** and **5F**, ECAD(+) MHs could form HBTOs. These data add important new information about the heterogeneity of hepatocytes. We also measured ECAD expression in cholangiocytes, supporting the hypothesis that ECAD expression in hepatocytes is important in forming adherence junctions with cholangiocytes (**Fig. S13**).

Reviewers' Comments:

Reviewer #1:

Remarks to the Author:

The reviewer thanks the authors for having considered all remarks; the manuscript is very significantly improved. The authors satisfactorily addressed the reviewer's comments #1 - 5. However, some questions remain unanswered, and a few additional minor points need to be addressed:

1. (comment #6 from 1st version) The reviewer clearly sees on the pictures how a bile canaliculus merges with a biliary lumen. But how is a "boundary" defined and how is the length of boundary determined? How does a "boundary" differ from a "connection"? Clear definitions would make the quantification of such connections more understandable.

2. (comment #7 from 1st version) Description of the methods is improved, but the number of replicates is still not clearly mentioned in several experiments. Dot plots would be more informative than bar graphs.

3. The data in Fig. S10A are confusing. According to the figure legend, only cholangiocytes were grown and no hepatocytes were added. While the qPCR results support that the cholangiocytes cultured alone maintain their characteristics in sandwich culture conditions, such data do not provide evidence that biliary marker expression is well maintained in hepatobiliary organoids. Also, the biliary marker/GAPDH ratio is similar in cholangiocyte cultures and in hepatobiliary organoids although the latter contain cholangiocytes and hepatocytes; the reviewer expects that the biliary marker/GAPDH ratio would be lower in the organoids because of dilution of the biliary marker by the hepatocytes.

The reviewer acknowledges that those data have been presented in the first version of the manuscript and that this comment could have been made while reviewing the first version.

4. The rebuttal letter states that the cholangiocytes do not display primary cilia. This information is important as it informs the reader about the maturity of the cholangiocytes; it should be mentioned in the discussion.

5. Lines 30-32 (abstract): The authors state: "This hepatobiliary organoid enabled us [...] to investigate the way in which the two types of epithelial cells establish functional connections." This sentence should be corrected as the way in which connections are established is not studied in the manuscript.

6. Lines 181-183. The authors conclude that: "The hepatobiliary connections in HBTOs promoted hepatic functions and contributed to maintaining these functions in the long term". This is overstated as we do not know whether it is the connection or any other event occurring in the coculture which improves hepatic function. This conclusion should be amended by stating, for instance, that the coculture conditions favor maturation of the hepatocytes (or a similar statement).

7. Lines 220-222. The following statement should also be tempered: "These results indicate that ECAD expression is required for hepatocytes to be involved in hepatobiliary connections with cholangiocytes, and that strong ECAD expression in SHs can explain their superiority over MHs for generating HBTOs. Such conclusion cannot be drawn in the absence of experiments targeting specifically E-cadherin's function.

8. Line 230: Coculture of human hepatocytes and tomato+ cholangiocytes is shown in Fig. 6A (3-4), not in fig. 6B.

9. The authors should clarify at what stage of organoid growth the properties of the cholangiocytes in the hepatobiliary organoids were quantified (Fig. 2; Fig. S10). The rebuttal letter mentions that this was performed after one month of co-culture, but the reviewer did not find this information in the manuscript.

10. While this manuscript was in revision, Bin Ramli and co-workers published a paper in *Gastroenterology* (2020, in press) showing production of human hepatic organoids in which bile flows from canaliculi into biliary cysts. This paper should be mentioned and discussed in the light of the authors' own findings.

Reviewer #2:

Remarks to the Author:

The manuscript has significantly improved, especially the presentation, and the authors have sufficiently addressed my comments. I have a few more minor comments, but I do not need to see another revision of the manuscript before publication.

Minor comments:

1) Fig. 5E P value is missing in the graph

2) Fig. 6B Tomato fluorescence is difficult to see and is punctate, is there a better example image?

3) Related to Comment 5: Would be informative to back up the answer with a Sytox stain or equivalent to assess viability at end of organoid culture

4) Related to Comment 8: For Table S1 it is unclear how many fields were examined for each condition. This should be made clear in the table or the table legend. Is there also a justification that the number of fields and cells counted is sufficient?

5) Related to Comment 9: "tdTomato" is used in the Methods section describing the mouse strain but "Tomato" is used in the text and figures; recommend using "tdTomato" throughout to be consistent.

Reviewer #3:

Remarks to the Author:

The manuscript has been improved in this revision, but there are still major deficiencies that have not been addressed. Publication can therefore not be recommended in its current form.

The key outstanding issues, relating to the original comments are described below.

Point 1: These issues have not been adequately addressed. There is no quantification for the data presented in Figs 1, 2, 3, and 6. Moreover, it is still not clear how many replicates the results of representative of, whether they were biological replicates or technical, how many lines.

Point 2: These data are not shown: the authors state that it is interesting but should demonstrate that at least one other non-hepatocyte non-cholangiocyte cell type does not form the connections when combined with the hepatocytes or cholangiocytes.

Point 4: The authors do not show ALP activity, which would be the basic enzymatic function expected from hepatocytes. Conversely, they show CYP1b1 activity, which is in fact not normally expressed in the liver and is not involved in liver metabolism. Similarly CYP3A4 is expressed in other tissues including intestine. It is also puzzling why CYPs could not be demonstrated in

organoids, but only shown when the HBTOs were separated. What is the explanation for this observation?

Point 7: This generic statement in the response is not sufficient. For each experiment in the paper, the number of replicates should be stated.

Reviewer #4:

Remarks to the Author:

I acknowledge that they have incorporated some figures (such as Sup Fig 1) to add clarity to the immunostainings and morphological characteristics of the cells. However, rather than answering the questions raised, the authors just moved some of the problematic figures to supplementary data.

I am still very concern about the methods sections; it is insufficient and the authors have not incorporated the suggestions made, including number of technical and biological replicas per experiment, bioinformatic analysis and insufficient description on the statistical analysis.

They have now included scale bars... but only in some of the figures.

When asked for statistics they just mention that there is no significant differences between groups, but they never showed the figures. They have also quantified number of connections but they do not do it over the course of time.

They have now included the missing data such as the expression of HNF4a, Cps1 and Tdo2, however there are statistical differences between the MH and the SH (Cps1 $p=0.017$ and Tdo2 $p=0.04$), so the level of expression of both cell types is not comparable. Unless they are considering statistical significance below 0.001... but again, they do not state in the methods what do they consider statistically significant, nor the number of technical/biological replicates.

Overall I do not think that this manuscript meets the quality criteria of Nature Comms.

Responses to reviewers' comments

Reviewer #1:

The reviewer thanks the authors for having considered all remarks; the manuscript is very significantly improved. The authors satisfactorily addressed the reviewer's comments #1 - 5. However, some questions remain unanswered, and a few additional minor points need to be addressed:

1. (comment #6 from 1st version) The reviewer clearly sees on the pictures how a bile canaliculus merges with a biliary lumen. But how is a "boundary" defined and how is the length of boundary determined? How does a "boundary" differ from a "connection"? Clear definitions would make the quantification of such connections more understandable.

The boundary is where hepatocytes contact cholangiocytes, and hepatocytes and cholangiocytes form a continuous luminal structure at the connection. We added an image to clearly demonstrate these definitions and the quantification in **Fig. S3**.

2. (comment #7 from 1st version) Description of the methods is improved, but the number of replicates is still not clearly mentioned in several experiments. Dot plots would be more informative than bar graphs.

We added dot plots to show how many fields in each sample we examined the hepatobiliary connections (**Figs. S3, S16, and S18**) and replaced bar graphs demonstrating qPCR results with dot plots to show the number of samples more clearly (**Figs. 5, S13, and S15**). (New Fig. S13 was Fig. S10 in the previous version.)

In addition, we added the number of culture wells used for specific assays in each section of experimental procedures.

3. The data in Fig. S10A are confusing. According to the figure legend, only cholangiocytes were grown and no hepatocytes were added. While the results support that the cholangiocytes cultured alone maintain their characteristics in sandwich culture conditions, such data do not provide evidence that biliary marker expression is well

maintained in hepatobiliary organoids. Also, the biliary marker/GAPDH ratio is similar in cholangiocyte cultures and in hepatobiliary organoids although the latter contain cholangiocytes and hepatocytes; the reviewer expects that the biliary marker/GAPDH ratio would be lower in the organoids because of dilution of the biliary marker by the hepatocytes.

The reviewer acknowledges that those data have been presented in the first version of the manuscript and that this comment could have been made while reviewing the first version.

The reason why the expression levels of cholangiocyte markers are similar in cholangiocyte culture and HBTOs may be that the proportion of cholangiocytes is much larger than that of hepatocytes in HBTOs. As we showed in **Fig.S11B**, cholangiocytes account for about 80 - 90% of cells in HBTOs.

4. The rebuttal letter states that the cholangiocytes do not display primary cilia. This information is important as it informs the reader about the maturity of the cholangiocytes; it should be mentioned in the discussion.

We added photos of transmission electron microscopy of HBTOs, which did not show primary cilia in the biliary cells forming tubules in the HBTOs (**Fig. S14**). We described this information also in the text (line 224 in page 15). Our current goal is to reconstitute the liver tissue structures, in particular generating the connection between the bile canaliculi and the biliary luminal network. The structural and functional properties of cholangiocytes in HBTOs are therefore sufficient. However, in future, it may be necessary to further optimize the culture conditions to induce further maturation of cholangiocytes in HBTOs.

5. Lines 30-32 (abstract): The authors state: "This hepatobiliary organoid enabled us [...] to investigate the way in which the two types of epithelial cells establish functional connections." This sentence should be corrected as the way in which connections are established is not studied in the manuscript.

We appreciate the reviewer's comment. In this revised version, we performed co-culture of hepatocytes with pancreatic duct cells in order to address the comment from **reviewer 3**, and found that they established a continuous luminal network, although we have not confirmed

whether the connection is functional by examining the transport of fluorescent dye or hepatocyte metabolites. This result shows a possibility that our culture methodology might be applicable for generating other types of epithelial connections. Therefore, we have decided to keep the sentence as it is.

6. Lines 181-183. The authors conclude that: "The hepatobiliary connections in HBTOs promoted hepatic functions and contributed to maintaining these functions in the long term". This is overstated as we do not know whether it is the connection or any other event occurring in the coculture which improves hepatic function. This conclusion should be amended by stating, for instance, that the coculture conditions favor maturation of the hepatocytes (or a similar statement).

Thank you for pointing out an important issue. We changed the sentence to “The generation of HBTOs promoted hepatic functions and contributed to maintaining these functions for a long term. We consider that the formation of hepatobiliary connections is correlated with improvement of hepatocyte functions, although we cannot exclude a possibility that co-culture conditions may favor the maturation of hepatocytes in HBTOs.”

7. Lines 220-222. The following statement should also be tempered: "These results indicate that ECAD expression is required for hepatocytes to be involved in hepatobiliary connections with cholangiocytes, and that strong ECAD expression in SHs can explain their superiority over MHs for generating HBTOs. Such conclusion cannot be drawn in the absence of experiments targeting specifically E-cadherin's function.

We thank the reviewer for pointing out a very important point. In the revised version, we changed the sentence to “These results indicate that MHs in ZONE1 and ZONE 2, and SHs, which share the cellular characteristics of strong ECAD expression, can be involved in hepatobiliary connections with cholangiocytes.”

8. Line 230: Coculture of human hepatocytes and tomato+ cholangiocytes is shown in Fig. 6A (3-4), not in fig. 6B.

We corrected the numbering of Figure 6A.

9. The authors should clarify at what stage of organoid growth the properties of the cholangiocytes in the hepatobiliary organoids were quantified (Fig. 2; Fig. S10). The rebuttal letter mentions that this was performed after one month of co-culture, but the reviewer did not find this information in the manuscript.

We added information about the culture duration in the text and the legend to **Fig. S13**.

10. While this manuscript was in revision, Bin Ramli and co-workers published a paper in *Gastroenterology* (2020, in press) showing production of human hepatic organoids in which bile flows from canaliculi into biliary cysts. This paper should be mentioned and discussed in the light of the authors' own findings.

We have quoted the *Gastroenterology* paper in the References, and described the comparison between their work and ours (Page 18 and 19): “Among those challenges, Ramli et al. established a liver organoid associated with biliary cysts (25). They demonstrated that bile acid analogue taken up by induced hepatocytes eventually accumulated in the cystic structures, although it is not clear how the BCs and the biliary lumen are connected. It is still difficult to confer mature hepatic functions on induced-hepatocytes. In contrast to previous reports (23-25), we used cholangiocytes and committed progenitors for hepatocytes to generate hepatobiliary organoids. Hepatocyte progenitors mature structurally and functionally in HBTOs in a similar way as MHs, and the apical domain of hepatocytes and cholangiocytes cooperatively construct the hepatobiliary connection.”

Reviewer #2 (Remarks to the Author):

The manuscript has significantly improved, especially the presentation, and the authors have sufficiently addressed my comments. I have a few more minor comments, but I do not need to see another revision of the manuscript before publication.

Minor comments:

1) Fig. 5E P value is missing in the graph

We added the *P* value to Fig. 5E.

2) Fig. 6B Tomato fluorescence is difficult to see and is punctate, is there a better example image?

We recognized that tdTomato signal in hybrid HBTOs is punctate in cholangiocytes after immunostaining procedure. The tdTomato signal is very different from that in images acquired for mouse HBTOs. Currently, we have no idea what causes this difference. We further tried to improve the quality of figure and replaced it with the previous one.

3) Related to Comment 5: Would be informative to back up the answer with a Sytox stain or equivalent to assess viability at end of organoid culture

We further examined the viability of hepatocytes in HBTOs with SYTOX green (**Fig. S17A**), demonstrating hepatocytes and cholangiocytes in HBTO are still alive at least at three weeks after MG-Col gel overlay.

4) Related to Comment 8: For Table S1 it is unclear how many fields were examined for each condition. This should be made clear in the table or the table legend. Is there also a justification that the number of fields and cells counted is sufficient?

We added information about the number of fields and cells we examined for conversion between hepatocytes and cholangiocytes in HBTOs in Supplementary Methods.

5) Related to Comment 9: "tdTomato" is used in the Methods section describing the mouse strain but "Tomato" is used in the text and figures; recommend using "tdTomato" throughout to be consistent.

In the revised version, we use tdTomato throughout the manuscript.

Reviewer #3 (Remarks to the Author):

The manuscript has been improved in this revision, but there are still major deficiencies that have not been addressed. Publication can therefore not be recommended in its current form.

The key outstanding issues, relating to the original comments are described below.

Point 1: These issues have not been adequately addressed. There is no quantification for the data presented in Figs 1, 2, 3, and 6. Moreover, it is still not clear how many replicates the results are representative of, whether they were biological replicates or technical, how many lines.

To reinforce the data shown in **Figs. 1, 2, 3, and 6**, we added new data in Supplementary Figures (**Figs. S3, S5, S6, S9, and S16**).

For Fig. 1, we added an image showing how we counted the number of hepatobiliary connections and evaluated the length of the boundary where hepatocytes contact cholangiocytes (**Fig. S3A**). We counted the number of hepatobiliary connections and evaluated the length of hepatocyte-cholangiocyte boundaries in more than eight different fields in five independent culture wells (**Fig. S3B**). The number of connections in HBTOs was determined as the average value of five samples and shown in the text.

For Fig. 2, three dimensional analysis was further performed for the immunofluorescence images with anti-EZN and CEACAM antibodies to evaluate the way in which hepatocytes and cholangiocytes form the hepatobiliary connection. About 60% of the connections consist of one hepatocyte and one cholangiocyte, whereas others consist of one hepatocyte and two or three cholangiocytes (**Fig. S6**).

For Fig. 3, CLF-positive luminal areas in biliary tubules were quantified using ImageJ. The data showed that CLF was transported from bile canaliculi to biliary tubules between 30 min and 6 hours (**Fig. S9**).

For Fig. 6, we counted the number of hepatobiliary connections and evaluated the length of hepatocyte-cholangiocyte boundaries in more than four different fields in four independent culture wells (**Fig. S16**). The number of connections in hybrid HBTOs was determined as the

average value of four samples and shown in the text.

Point 2: These data are not shown: the authors state that it is interesting but should demonstrate that at least one other non-hepatocyte non-cholangiocyte cell type does not form the connections when combined with the hepatocytes or cholangiocytes.

We performed the co-culture of mouse pancreatic duct cells with small hepatocytes. We found that they generated a continuous luminal network. (**Fig. S20**). Although we have not certified whether the duct and bile canaliculus are functionally connected by examining the transport of fluorescent dye., the result suggests a possibility that our culture method might be applicable to connect other tubular structures for generating functional epithelial tissues *ex vivo*.

Point 4: The authors do not show ALP activity, which would be the basic enzymatic function expected from hepatocytes. Conversely, they show CYP1b1 activity, which is in fact not normally expressed in the liver and is not involved in liver metabolism. Similarly CYP3A4 is expressed in other tissues including intestine. It is also puzzling why CYPs could not be demonstrated in organoids, but only shown when the HBTOs were separated. What is the explanation for this observation?

We measured ALP activity and showed the data in **Fig. S17B**. In consistent with that ALP is normally localized in bile canalicular membrane, ALP activity was detected after treating HBTOs with troglitazone, a hepatotoxin.

We really appreciate the reviewer to point out our misunderstanding what type of CYP activity we measured with Luciferin-CEE provided in P450-Glo™ CYP1B1 Assay System (Promega). As the reviewer pointed out, hepatocytes express CYP1A1 but not CYP1B1, we actually measured Cyp1A1 activity with this substrate. We changed the labeling from Cyp1B1 to CYP1A1 in **Fig 4**. We also explained that CYP1A1 can be measured using P450-Glo™ CYP1B1 Assay System in page 30.

In this revised manuscript, we further tried immunostaining of HBTOs with another antibody against human CYP3A4 commercially supplied from Abcam, which is reported to be cross-reactive to mouse CYP3A protein. The result showed that CYP3A protein is expressed in hepatocytes in HBTOs (**Fig. S12B**). We consider this antibody detects mouse CYP3A11, whose

mRNA expression is detected in HBTOs (**Fig. S11**) and its sequence shows 69% homology to human CYP3A4. However, since mice have other paralogs, we labeled CYP3A for the signal detected with this antibody in panels of **Fig. S12**.

Point 7: This generic statement in the response is not sufficient. For each experiment in the paper, the number of replicates should be stated.

We carefully checked the manuscript and added information about the number of culture wells used for quantifying CYP activities, ALB secretion, and the number of hepatobiliary connections. We also indicated the number of samples used in PCR analyses. We show representative images in Fig. 3, and we indicated how many times we examined the transport of CMFDA, CLF and bilirubin. This information is described in the Materials and Methods.

Reviewer #4 (Remarks to the Author):

I acknowledge that they have incorporated some figures (such as Sup Fig 1) to add clarity to the immunostainings and morphological characteristics of the cells. However, rather than answering the questions raised, the authors just moved some of the problematic figures to supplementary data.

1. I am still very concern about the methods sections; it is insufficient and the authors have not incorporated the suggestions made, including number of technical and biological replicas per experiment, bioinformatic analysis and insufficient description on the statistical analysis. They have now included scale bars... but only in some of the figures.

We carefully checked the manuscript and added information about the number of culture wells used for quantifying CYP activities, ALB secretion, and the number of hepatobiliary connections. We also indicated the number of samples used in PCR analyses. We show representative images in Fig. 3, and we indicated how many times we examined the transport of CMFDA, CLF and bilirubin. This information is described in the Materials and Methods.

For RNA-seq analysis, we added a description of the data collection and analysis in Supplemental Methods. We have deposited the data in the Gene Expression Omnibus, and provided the accession number in the Data Availability section.

We carefully checked the figures and added scale bars to all Figures.

We explain more precisely about the statistical analysis in Materials and Methods. Unpaired two-tailed student *t*-tests were performed for the data of CYP1A1 & CYP3A4 activity, ALB secretion, and the counting of hepatobiliary connections using Microsoft Excel. Paired two-tailed student *t*-tests were performed for quantitative PCR data, comparing the expression of hepatocyte marker genes in mature hepatocytes (MH) and small hepatocytes (SH). When the result of *t*-test had a $P < 0.05$, we concluded that the difference between the two groups was statistically significant.

We also added information how many culture samples were examined. The numbers of culture samples used for each analysis were as follows: $n = 5$ (the number of hepatobiliary connection), $n = 5$ (ALB secretion), $n = 3$ (CYP activities), $n = 8$ (qPCR examining *Cdh1* and *Cldn2* expression in SHs and MHs), $n = 4$ (qPCR for other hepatocyte markers). For technical replicates, duplicates of each sample were generated, and the average values were used in statistical analyses for data of CYP activities and qPCR.

2. When asked for statistics they just mention that there is no significant differences between groups, but they never showed the figures. They have also quantified number of connections but they do not do it over the course of time.

We added images to explain our quantification of the number of hepatobiliary connections in HBTOs. For **Fig. 1**, we added an image showing how we counted the number of hepatobiliary connections and evaluated the length of the boundary where hepatocytes contact cholangiocytes (**Fig. S3A**). We counted the number of hepatobiliary connections and evaluated the length of hepatocyte-cholangiocyte boundaries in more than eight different fields in five independent culture wells (**Fig. S3B**). The number of connections in HBTOs was determined as the average value of five samples and is shown in the text. For **Fig. 6**, we counted the number of hepatobiliary connections and the length of hepatocyte-cholangiocyte boundaries in more than four different fields in four independent cultures consisting of hCLiPs and mouse cholangiocytes (**Fig. S16**). The number of connections in hybrid HBTOs was determined as the average value of four samples and shown in the text.

In the revised version, we further examined the hepatobiliary connections in HBTOs at 1W and 4W after MG-Col gel overlay. The data demonstrated that the number of connections is similar

between one and four weeks, suggesting that the connection is established at the early stage of morphogenesis in co-culture. On the other hand, the length of the bile canaliculi grew between one and four weeks. We show these data in **Fig. S18**.

In addition to the number of hepatobiliary connections, we also performed quantification for the data shown in **Figs. 2 and 3**. For **Fig. 2**, immunostaining with anti-EZN and anti-CEACAM antibodies and three dimensional analyses were further performed to evaluate how hepatocytes and cholangiocytes form hepatobiliary connections. About 60% of the connections consist of one hepatocyte and one cholangiocyte, whereas others consist of one hepatocyte and two or three cholangiocytes (**Fig. S6**). For **Fig. 3**, CLF-positive luminal areas in biliary tubules were quantified using ImageJ. The data showed that CLF taken up by hepatocytes was transported to biliary tubules between 30 min and 6 hours (**Fig. S9**).

3. They have now included the missing data such as the expression of HNF4a, Cps1 and Tdo2, however there are statistical differences between the MH and the SH (Cps1 p=0.017 and Tdo2 p=0.04), so the level of expression of both cell types is not comparable. Unless they are considering statistical significance below 0.001... but again, they do not state in the methods what do they consider statistically significant, nor the number of technical/biological replicates.

We appreciate the reviewer identifying that the expression of *Cps1* and *Tdo2* are different between MH and SH. We further performed qPCR analyses with technical replicates, and show the revised data in **Fig. S15**. Since we consider that $P < 0.05$ shows statistically significant difference, we changed the legend to **Fig. S15A**, as "*Hnf4a* is expressed more and *Tdo2* is expressed less, in SHs as compared with those in MHs. *Cps1* is expressed significantly more in SHs.".

Overall I do not think that this manuscript meets the quality criteria of Nature Comms.

With new data, rewriting the manuscript, and further statistical analyses, we believe our manuscript now meets the quality criteria of Nature Communications.

Reviewers' Comments:

Reviewer #3:

Remarks to the Author:

The Authors have addressed most of my concerns and suggestions. However, as noted by other reviewers also, the number of experiments and replicates performed are still not entirely clear. Some data have been added to two of the supplementary figures, showing 4 or 5 replicates for those experiments. The authors should add a clear statement, for every main figure in the manuscript about the number of replicates. For example, the figure legends should all have a statement such as "Images are representative of X independent experiments".

Reviewer #4:

None

Reviewer #3 (Remarks to the Author):

The Authors have addressed most of my concerns and suggestions. However, as noted by other reviewers also, the number of experiments and replicates performed are still not entirely clear. Some data have been added to two of the supplementary figures, showing 4 or 5 replicates for those experiments. The authors should add a clear statement, for every main figure in the manuscript about the number of replicates. For example, the figure legends should all have a statement such as "Images are representative of X independent experiments".

We appreciate the comments from the reviewer.

We added the following sentences in legends. "The experiment (or immunostaining) was repeated X times, independently. Three (or two) fields were examined in each well and the representative images are shown in this figure". X is fifty (Fig. 1b), five (Fig. 1c), four (Fig. 6a and 6b), three (Fig. 2a and 2b, Fig. 3a, 3b and 3c, Fig. 5a, 5e), or two (Fig. 4d, Fig. 5b, Fig. 6c).

For FACS data, we added a sentence "The FACS plot is representative of X independent analyses". X is three (Supplementary Fig. 16), and two (Fig. 5d, Supplementary 11b, 20, and 21a).

We also provided the number of replicates in Supplementary Figures. X is five (Supplementary Fig. 3a), four (Supplementary Fig. 17), three (Supplementary Fig. 1a, 1b, 7, 16, 19a) or two (Supplementary Fig. 2, 4a, 4b, 5, 8, 11a, 12a, 12b, 13b, 13c, 14, 18a, 19a, 19c, 21b).